# Corrosion Behaviors of Heat-Resisting Alloys in High Temperature Carbon Dioxide

**DOI:** 10.3390/ma15041331

**Published:** 2022-02-11

**Authors:** Liujie Yang, Hongchen Qian, Wenjun Kuang

**Affiliations:** Center for Advancing Materials Performance from the Nanoscale (CAMP-Nano), State Key Laboratory for Mechanical Behavior of Materials, Xi’an Jiaotong University, Xi’an 710049, China; yanglj1995@stu.xjtu.edu.cn (L.Y.); qianhongchen2021@163.com (H.Q.)

**Keywords:** high temperature CO_2_ corrosion, heat-resisting alloys, oxidation, carburization, stress corrosion

## Abstract

The supercritical carbon dioxide Brayton cycle is a promising power conversion option for green energies, such as solar power and nuclear reactors. The material challenge is a tremendous obstacle for the reliable operation of such a cycle system. A large body of research indicates that high-temperature corrosion of heat-resisting alloys by CO_2_ results in severe oxidation and, in many cases, concurrent internal carburization. This paper mainly reviews the oxidation behavior, carburization behavior and stress corrosion behavior of heat-resisting alloys in high temperature CO_2_. Specifically, the main factors affecting the oxidation behavior of heat-resistant alloys, such as environmental parameters, surface condition and gaseous impurity, are discussed. Then, carburization is explored, especially the driving force of carburization and the consequences of carburization. Subsequently, the effects of the environmental parameters, alloy type and different oxide layers on the carburizing behavior are comprehensively reviewed. Finally, the effects of corrosion on the mechanical behavior and stress corrosion cracking behavior of heat-resisting alloys are also summarized. The corrosion performances of heat-resisting alloys in high temperature CO_2_ are systematically analyzed, and new scopes are proposed for future material research. The information provided in this work is valuable for the development of structural material for the supercritical carbon dioxide Brayton cycle.

## 1. Introduction

With the fast development of the global economy, the excessive consumption of fossil energy has seriously polluted the environment and sabotaged the ecosystem. In order to sustain the usage of fossil fuel and improve the global environment, it is imperative to maximize the efficiency of fossil fuel and increase the use of clean energy, such as solar and nuclear power.

Improving the efficiency of an energy conversion system is an important approach to diminish the consumption of fossil fuel. The supercritical carbon dioxide (S-CO_2_) Brayton cycle is being considered as a promising power conversion system, which uses the S-CO_2_ as working fluids. This is because the S-CO_2_ Brayton cycle has higher efficiency of energy conversion, lower cost, lower noise and a more compact design [1,2]. As shown in Figure 1 [3], the current average efficiency is about 34% and the potential efficiency of the S-CO_2_ Brayton cycle is close to 50%, which exceeds the efficiency of the high temperature helium Brayton cycle and the traditional steam Rankine cycle. Such high efficiency can greatly reduce both fossil energy consumption and the cost of electricity. Moreover, the critical point of CO_2_ is 7.383 MPa and 31.06 °C, where the distinction between liquid and gas disappears (Figure 2a). This implies that S-CO_2_ fluids have lower viscosity (like gas) and higher density (like liquid). The compressibility of CO_2_ changes drastically near the critical point. Figure 2b displays the compressibility factor of CO_2_ in the vicinity of the critical point, wherein the compressibility factor is defined as the molecular volumetric ratio of a fluid compared with an ideal gas. It describes how much the fluid behaves like ideal gas. When this factor is unity, the fluid behaves very close to an ideal gas. If the compressibility factor is zero, it can be considered as an incompressible fluid. The compressibility factor of CO_2_ is about 0.2 to 0.5, indicating that the compression work can be greatly reduced in a small temperature range close to the critical point. This behavior is one of the main reasons why the S-CO_2_ Brayton cycle has the potential to improve the efficiency of the power conversion.

Figure 3 shows a schematic diagram of the closed Brayton cycle system, which is mainly composed of a compressor, recuperator, turbine, precooler and heat source, such as nuclear energy [4,5], solar energy [6,7], geothermal energy [8] or waste heat [9]. Hence, it can be used in a nuclear power plant and thermal power plant [10,11]. The key components, such as the compact-type intermediate heat exchanger, turbine and some pipes, are directly exposed to the high temperature (650–750 °C) and high pressure (20–30 MPa) S-CO_2_ environment [12,13]. Although the corrosiveness of CO_2_ is weak at a low temperature, it can still induce severe corrosion of materials at a high temperature, as in the Brayton cycle system. The corrosion of structural material in high temperature high pressure CO_2_ is a major issue that hinders the practical applications of this cycle system. In addition to static corrosion, the key components are also subject to mechanical loading in service, which would accelerate the corrosion process. Thus, the biggest challenge for the structural materials used in S-CO_2_ is to maintain the structural integrity and load bearing capacity under the coupling effects of corrosion and stress during the lifetime, which is around 20 years or longer.

So far, there is no alloy specifically designed for the S-CO_2_ Brayton cycle system. As early as the 1960s, researchers began to study the corrosion of steels in gas-cooled nuclear reactors, in which CO_2_ is the main coolant [14,15,16]. More recently, the corrosion behaviors of many heat-resisting alloys, such as ferritic–martensitic steels (FMs) [17,18,19,20,21,22,23], austenitic steels [19,22,24,25,26], nickel-base alloys [12,27,28,29] and other new-type alloys [30,31,32], in high temperature CO_2_ have also been studied extensively. The results show that the corrosion forms of heat-resistant alloys in a high temperature CO_2_ environment are mainly oxidation and carburization. It seems that the corrosion of metallic material in high temperature CO_2_ is inevitable as quite low oxygen activity and carbon activity can still induce oxidation and carburization [33]. The formation of thick oxides and internal carbides can impair the heat-transfer capability and degrade the mechanical property [29,34]. When the surface oxide layer is breached and loses its protectiveness, breakaway corrosion occurs, during which the oxidation and carburization of the alloy are greatly accelerated. More importantly, the load-bearing capacity of alloys can be significantly compromised in the process of stress corrosion. These degradations can significantly shorten the service life of a structural alloy. It is essential to understand the corrosion behaviors of structural alloys in high temperature CO_2_ in order to predict or extend the lifetimes of alloys and ensure the safe operation of a component. Despite great research efforts, some challenges in this field remain unresolved. For example, the crack initiation and propagation behaviors of heat-resistant alloys in high temperature CO_2_, which greatly affect the load-bearing capacity of component, are still poorly understood. The intergranular corrosion behavior of heat-resistant alloys in high temperature CO_2_ is also a critical issue. It is of great importance to understand the preferential intergranular corrosion behavior, which is necessary to predict and improve the intergranular stress corrosion cracking performance of alloys. The purpose of this paper is to offer a brief overview of the research works on the corrosion behavior of candidate metallic materials in high temperature CO_2_. Specifically, the different types of corrosion behaviors (oxidation, carburization and stress corrosion) of heat-resisting alloys in high temperature CO_2_ will be reviewed, and the effects of the major influencing factors will be covered. Afterwards, the prospects for future work will be discussed.

## 2. Oxidation

### 2.1. Corrosion Kinetics

Weight gain and thickness of oxide layer are generally measured to assess the oxidation rates of alloys in high temperature CO_2_ [35]. Two rate constants can be acquired from the measurements. One is the $K_{w}$, which is related to mass change of alloys. Another is the $K_{P}$, which is calculated from the thickness of oxide film. They are both parabolic rate constants. The smaller the calculated value of $K_{w}$ or $K_{P}$ is, the higher the corrosion resistance of the alloy is. The related expression is as follows [33,35]:(1)$${\left(\frac{\Delta W}{A}\right)}^{2}=2\;k_{w}t$$
(2)$$X^{2}=2\;k_{P}t$$
where ΔW denotes the mass change of the alloy, *A* is the surface area of the alloy, *X* represents the thickness of the oxide layer and *t* is the time.

Figure 4a shows the schematic diagram of typical corrosion kinetics of 9-12Cr FMs, austenitic steels and nickel-base alloys. These results follow a parabolic corrosion law, indicating that the growth rate of oxide is controlled by diffusion. The 9-12Cr FMs display the largest oxidation rates, while others exhibit relatively small and similar oxidation rates. This is because the surface of 9-12Cr FMs is covered by Fe-rich oxide, while the surfaces of austenitic steels and nickel-base alloys possess protective chromia layer (Figure 4b–d), which may be the reason for the lower corrosion kinetics of nickel-base alloys and austenitic steels. However, although austenitic steels show similar corrosion kinetics to nickel-base alloys, it has been reported that breakaway corrosion occurs more easily in austenitic steels compared to nickel-base alloys with similar Cr content [36]. In other words, the protective chromia scale is interrupted by the subsequently non-protective Fe-rich oxide nodules on the surface. This will give rise to a sharp increase in the corrosion kinetics of austenitic steels (Figure 4a,c). This also suggests that the corrosion kinetics of the alloy is closely related to the corrosion products formed on the surface of the alloy. Therefore, more attention should be paid to the features of oxide structure of conventional alloys, such as FMs, austenitic steels and nickel-base alloys.

### 2.2. FM Steel

The 9-12Cr FMs are considered as possible candidate structural materials in high temperature parts of the cycle because they have low growth stress of oxide, low thermal stress, good heat transfer properties and good mechanical properties. It has been reported that 9-12Cr ferritic steels and 9-12Cr martensitic steels possessed similar oxidation behaviors in high temperature CO_2_ [17,22]. As shown in Figure 5a [17], 9Cr ferritic steel forms a duplex oxide scale composed of a porous hematite/magnetite layer and an almost as thick inner Fe–Cr spinel oxide layer in 550 °C/1 bar CO_2_ after 350 h. Hematite, which becomes more adherent on the matrix than magnetite, will form on the outermost magnetite if there are some obstacles, such as gaps or pores, restricting the outward diffusion of Fe during corrosion [48,49]. The magnetite layer is composed of columnar grains, and the hematite layer is made of small equiaxed grains. Due to the compressive stress caused by oxide growth, the outer hematite may spall. In contrast, the magnetite usually cracks because of the tensile stress caused by oxide growth [50]. Additionally, it has been shown that the inner Fe–Cr spinel layer was non-stoichiometric and usually described as Fe_3-X_Cr_X_O_4_ [17,41,43] or even more generally as Fe_3_O_4_ + FeCr_2_O_4_ or (Fe,Cr)_3_O_4_ [37,51]. Recently, it has been reported that the Fe–Cr spinel oxide layer was comprised of small equiaxed grains, of which the stoichiometry evolved from Fe_2.6_Cr_0.4_O_4_ to Fe_1.9_Cr_1.1_O_4_ in 550 °C/1 bar CO_2_ gas [52].

As shown in Figure 5b, an internal oxidation zone (IOZ), which is filled with Cr-rich oxide particles, is formed between Fe–Cr spinel oxide and Cr-rich carbides. These carbides are mainly caused by carburization, which will be described in detail in the next section. Moreover, the corrosion behavior of 12Cr martensitic steel in 200 bar S-CO_2_ was investigated in the temperature range of 400 to 600 °C for 8000 h [40,53]. In this work, the exfoliation of the oxide layer was observed. The results displayed that carburization was the main reason for the spalling of the oxide layer in this martensitic steel. This is because the carbon deposition in the porous oxide layer, such as magnetite layer, will lead to the spallation of the oxide layer by generation of high stress. Furthermore, it is also indicated that carbon accumulated in the inner oxide scale during carburizing can improve the hardness by the formation of brittle carbides, which will promote the peeling of the protective oxide layer [54]. Therefore, the exfoliation of the oxide layer is mainly related to the carburization in 9-12Cr FMs.

F. Rouillard et al. [43,44,55] have systematically studied the corrosion behavior of 9Cr1Mo steel in CO_2_ at 550 °C/1 bar and proposed a mechanism of void-induced duplex oxide formation known as “available space model”. The “available space model” has two important viewpoints. On the one hand, the growth of the inner oxide occurs at the oxide/metal interface, and the growth of the outer oxide occurs at the oxide/medium (original surface of the alloy) interface. On the other hand, the oxidant species diffuse in molecular form through fast diffusion paths, such as pores and microchannels, to the metal/oxide interface. The theory states that a magnetite layer grows outwards, injecting iron vacancies, and an inner Fe–Cr spinel oxide layer grows inwards inside the volume created by iron vacancies condensation. Thus, when the freshly formed Fe–Cr spinel oxide fills the nanopores created by iron vacancies condensation, the fast diffusion paths will be blocked. Only when the nanopores are formed again, the fast diffusion paths can be generated and the fresh Fe–Cr spinel oxides can be formed [56]. In this theory, the Cr element never moves. Meanwhile, the experimental result displayed that the volumetric concentration of Cr in Fe–Cr spinel oxide was roughly the same as that in steel [43]. Hence, it could be considered that the Cr element hardly moved during the oxidation process in this test. This shows that the “available space model” is consistent with the experimental results. Another interesting feature, according to this theory, is that the thickness ratio of the outer oxide layer to the inner oxide layer is a constant regardless of the time and temperature. It has been reported that the thickness ratio was 1.4 [43,52], and it further indicated that the corrosion kinetics of the outer scale was closely associated with that of the inner scale.

In addition to 9-12Cr FMs, the oxidation behavior of high-chromium FMs Fe20Cr was also studied in 650 °C/1 bar Ar-20%CO_2_ gas for 120 h [33]. As shown in Figure 6, the FMs Fe20Cr forms a thin Cr_2_O_3_ layer interrupted by many Fe-rich oxide nodules. Compared with the 9-12Cr FMs, the matrix of FMs Fe20Cr is partially protected, which is associated with the high Cr content. Therefore, increasing the Cr content can slow down the oxidation rate of the FMs. Another report also suggested that the surface of commercial FMs 430(18Cr0.5Ni0.8Mn0.5Si) and FMs 630(17Cr4Ni0.6Mn0.3Si) could still maintain the growth of the chromia scale after corrosion in 650 °C/20 MPa CO_2_ for 1000 h [19]. This relatively superior oxidation behavior may be related to trace elements, which are present in commercial FMs but not in FMs Fe20Cr model alloy. Moreover, compared with austenitic stainless steels (310SS, Al-6XN, 800H) and nickel-base alloys (Hay230, 625), the corrosion resistance of oxide dispersion strengthening FMs PM2000 (Fe19Cr5.5Al) was significantly improved under 650 °C/20 MPa S-CO_2_ environment [57]. This may be because the oxide dispersion strengthening (ODS) FMs PM2000 possesses a very dense and protective alumina scale, which is more oxidation-resistant than chromia. Therefore, the ODS Fe–Cr–Al alloy is expected to be a candidate material for the key components of a S-CO_2_ Brayton cycle system. Additionally, the excellent oxidation resistance may also be related to the dispersed oxides in the internal alloy. Perhaps, the strengthening method can also be applied to other materials, such as nickel-base alloys or austenitic steels, which may produce unexpected effects on oxidation resistance.

### 2.3. Austenitic Steel

Austenitic steels have been widely used as structural materials for core components due to their high heat conductivity, good high-temperature mechanical properties, high resistance to corrosion and relatively low cost. When the temperature was higher than 550 °C in a CO_2_ environment, it was demonstrated that the austenitic steel showed higher oxidation resistance than the 9-12Cr FMs [22,26,38,39] as it formed a protective chromia scale on the surface (Figure 7a) [19]. After all, the oxidation resistance of austenitic steels relies on their ability to rapidly form a uniform chromia scale on the surface as this scale is able to slow down the oxidation of the substrate. Although it has good corrosion resistance, it does not resist the formation of large amounts of carbides caused by carburizing (Figure 4c). Moreover, the spallation of the oxide layer was also observed in austenitic steel 347H, which destroyed the protective scale and accelerated the consumption of Cr, resulting in the formation of a severe Cr-depleted zone. The cause of the peeling of austenitic steel is different from that in 9-12Cr FMs. Compared with the ferritic/martensitic structure, the austenitic structure has a lower diffusion coefficient, larger growth stress and thermal stress during oxidation, which is an important cause of oxidation spallation.

With the reaction time extending, the surface of austenitic steel 347H (Fe9Ni18Cr) is transformed from a layer of Cr_2_O_3_ film to a layer of chromia containing many Fe-rich oxide nodules, while the nodules mainly form on the location of spallation or severe Cr-depletion zone (Figure 7b) [19]. This transition from a protective Cr_2_O_3_ scale to a non-protective Fe-rich oxide film is known as breakaway corrosion [58,59]. This behavior usually occurs in austenitic steels (Figure 7) and high-Cr FMs (Figure 6). It has also been reported that austenitic steel 800H (32Ni21Cr) could still maintain an intact chromia layer on the surface after 3000 h in 650 °C/20 MPa S-CO_2_ condition [25]. This indicates that the breakaway corrosion of austenitic steels is highly dependent on the alloy composition, especially Cr and Ni content. Hence, 9-12Cr FMs are also quite susceptible to breakaway corrosion due to low Cr content. Another work has reported that the alumina-forming austenitic steel gradually suffered breakaway corrosion with increasing temperature in the 450–650 °C/20 MPa S-CO_2_ condition [60], and the cracking of the oxide layer led to rapid breakaway corrosion of austenitic steel 316LN in 650 °C/20 MPa S-CO_2_ environment [19]. Additionally, other results indicated that pressure [41] or carburization [61] could also accelerate the occurrence of breakaway corrosion. In summary, the breakaway oxidation of an alloy is related to alloy composition, temperature, pressure and carburization. This behavior can significantly increase the rates of oxidation and carburization. Moreover, the breakaway corrosion, which is prevalent on austenitic steel and FMs, is also observed on nickel-base alloys at present [62]. Therefore, this issue needs to be resolved for the application of conventional alloys in a S-CO_2_ Brayton cycle system.

### 2.4. Nickel-Base Alloy

Nickel-base alloy is a vital candidate structural material for the thin-walled heat exchanger and the turbine in the S-CO_2_ Brayton cycle system because it has good mechanical properties and high corrosion resistance at high temperature. As shown in Figure 8a,b [46], a uniformly thin and protective Cr_2_O_3_ scale formed on nickel-base alloy 625 after 500 h exposure to S-CO_2_ at 650 °C/20 MPa. This may be one of the reasons why nickel-base alloy has excellent corrosion resistance. However, the spallation of the oxide layer on the surface of nickel-base alloy 625 could also be observed in Figure 8c and d [46]. Unlike austenitic steel, new Cr-rich oxides instead of Fe-rich oxides were formed on the spallation zone. This may be attributed to the very low Fe content and high Cr content of the nickel-base alloys or the lower diffusion rate of Fe in nickel-base alloys than in austenitic steels. This result indicates that the oxide film on nickel-base alloy has a self-healing ability, further confirming the superior corrosion resistance of nickel-base alloy. Moreover, it has also been indicated that when the Cr contents of nickel-base alloy and austenitic steel were similar, the chromia scale and iron-rich oxide layer were formed on nickel-base alloy and austenitic steel, respectively, after 1500 h at 650 °C/20 MPa [46]. Other studies have revealed that the cyclic oxidation performances of nickel-base alloys were better than those of austenitic steels [7,63] and the oxidation rate of the alloy decreased with increasing Cr and Ni content in 550 °C/20 MPa S-CO_2_ condition [64]. Overall, the nickel-base alloys exhibit better corrosion resistance than austenitic steels in a high temperature S-CO_2_ environment.

The mainly protective corrosion product of nickel-base alloy in high temperature CO_2_ is a chromia layer or alumina layer. For chromia-forming nickel-base alloys, such as alloy 690 or alloy 600, they rely on the chromia layer to prevent oxygen from further attacking the alloy in high temperature CO_2_ [65]. Moreover, due to the rapid diffusion of Mn in the chromia layer, some Mn–Cr spinel oxides were formed on the surface of the chromia layer in nickel-base alloys (Figure 4d). It was reported that the Mn–Cr spinel oxides grew slower than the single chromia, which slowed down the consumption of Cr [66]. This is beneficial to the corrosion resistance of nickel-base alloys. Meanwhile, it was found that the (Al, Si, Ti)-rich oxides could be observed at the chromia/matrix interface in nickel-base alloys (Figure 4d). It was also indicated that these trace elements could reduce the oxidation rates of nickel-base alloys to a certain extent [67]. Therefore, the trace elements also play an important role in the corrosion resistance of nickel-base alloys. For alumina-forming alloys, such as alloy 214, it has been reported that this alloy was covered by a thin alumina layer that drastically improved the corrosion resistance in S-CO_2_ at 750 °C/20 MPa for exposure durations up to 500 h [12]. In addition, it has also been reported that alumina-forming alloys displayed thinner corrosion products at higher temperatures (650–750 °C) as alumina formed faster [12]. Therefore, the use of alumina-forming nickel-base alloys may be an option at higher temperatures, typically higher than 750 °C. Interestingly, carburization at the oxide/metal interface was observed in chromia-forming nickel-base alloy 600 and alloy 690 (Figure 4d) but not in alumina-forming nickel-base alloy 214 [12]. Hence, the alumina layer is a better barrier against carburization than the chromia layer [57,68]. Although the chromia-forming nickel-base alloy has superior oxidation resistance, it is also subject to carbon deposition or carburization at the oxide/metal interface [28,46]. Such a carburization, which weakens the bonding between the oxide and matrix, will increase the probability of the exfoliation of the oxide layer. Surprisingly, no carbides are formed inside nickel-base alloys, indicating that nickel-base alloys have excellent carburization resistance [29,65]. Thus, alumina-forming nickel-base alloys may be a candidate material for the S-CO_2_ Brayton cycle system.

### 2.5. Effects of Environmental Parameters

There are many factors that affect the oxidation rate of an alloy, such as pressure, temperature and gas flowing rate. At present, there is a great deal of information about the influence of temperature and pressure on the corrosion behaviors of 9-12Cr FMs, austenitic steels and nickel-base alloys in a CO_2_ environment. As shown in Figure 9, the increase in temperature would accelerate the oxidation rates of FMs (P122), austenitic steel (800HT) and nickel-base alloy (600 and 690) [17,29]. This is because higher temperature results in faster molecular/atomic diffusion, leading to more severe corrosion. Changes in CO_2_ pressure have different effects on the corrosion behaviors of different alloys. For the austenitic steel 800HT and nickel-base alloys 600 and 690, the higher the pressure is, the greater the mass change is. This phenomenon may be induced by the enhancement of the inward diffusion of the oxidant through the grain boundaries or defects in the oxide layer under high pressure [65]. In addition, it has also been demonstrated that the pressure had little effect on the mass change or reaction rate of a group of structural alloys containing Cr or Al after 500 h exposures in 750 °C/(1–300 bar) CO_2_ [69]. Surprisingly, for FMs F91, austenitic 310HN and nickel-base alloy 617B, the beneficial effect of increasing pressure was observed [48,70]. This is due to the fact that FMs F91 forms a porous oxide structure with a few cracks in the case of 0.1 MPa CO_2_, while a uniform and dense oxide structure is formed in the case of 5 MPa CO_2_ [48]. Holes or cracks can provide fast access to the metal/oxide interface, accelerating the diffusion of gas and ultimately leading to a quick oxidation rate [71,72]. In general, it suggests that increasing the pressure may reduce the formation of porosity and increase the adhesion of the alloy.

Interestingly, a study has indicated that the oxidation rate of FMs T91 alloy was hardly affected by pressure [17]. Instead, it was found that the oxidation rate of FMs T91 alloy was influenced by the gas flowing rate. The mass change obtained in the static CO_2_ environment was only half of that obtained in the flowing CO_2_ environment [17]. In static CO_2_, there was no hematite formed on the top of FMs T91, which was present in flowing CO_2_ (such as Figure 5). The reason for the different results may be due to the difference in oxygen activity. Therefore, the flowing rate of gas is also extremely important for corrosion. However, the effect of the gas flowing rate on the oxidation behavior of alloys has not been studied in detail, which is of great significance for the application of the S-CO_2_ Brayton cycle system, especially for heat transfer.

Some researchers have used Fe–Cr and Ni–Cr binary alloys for corrosion tests in Ar-20%CO_2_ gas. As shown in Figure 10, the oxidation rates of Fe9Cr, Ni10Cr and Ni20Cr binary model alloys increased with the increase in temperature. This result is the same as that observed on commercial alloys in Figure 9. However, as the temperature increased, the oxidation rates of Fe20Cr, Ni25Cr and Ni30Cr binary model alloys unexpectedly decreased [47,73,74]. The different result is closely related to the diffusion rate of Cr and the Cr content of the alloy. For alloys with high Cr content, the outward diffusion rate of Cr is greatly improved by higher temperature, which promotes the formation of a protective external Cr_2_O_3_ scale to slow down the oxidation rate. For alloys with low Cr content, such as Fe9Cr or Ni10Cr, although the outward diffusion of Cr can be enhanced by increasing temperature, the alloy cannot provide enough Cr to form a protective Cr_2_O_3_ film, resulting in the formation of a non-protective Fe-rich or Ni-rich oxide layer on the surface. In other words, even if the temperature can promote the rapid formation of a chromia scale, the alloy itself needs to have enough Cr content to play a protective role. Therefore, the more Cr the alloy contains, the higher the corrosion resistance it has. Interestingly, an increase in temperature does not always improve the oxidation rate of the alloy in a short time.

### 2.6. Effects of Surface Treatments

As shown in Figure 11a and b [75], when the surface had been polished, a little thin chromia scale with many thick Fe-rich oxide nodules could form on the surface of austenitic steel 347H in 550 °C/20 MPa S-CO_2_ for 1500 h. When the surface had been ground by 600-grit silicon carbide sandpaper, a continuously thin chromia layer with very small Fe-rich oxide nodules was formed on the surface (Figure 11c,d) [75]. Further comparison reveals that the increase in surface roughness slows down the oxidation rate of austenitic steel 347H. This is because the deformed layer of the surface can promote the formation of protective Cr_2_O_3_ scale by promoting the outward diffusion of reactive solute elements (such as Cr). Surface polishing can remove hardened or deformed layers on the surface. Therefore, the oxidation of the alloy can be strongly influenced by the surface condition. The possible reason for the improvement of corrosion resistance of austenitic steel 347H is that the stress-induced martensite transition caused by the surface treatment will reduce the grain size in the surface zone to produce a high density of the grain boundary. This increases the channel for the rapid diffusion of the reactive solute elements, further accelerating the formation of the protective scale. This phenomenon is known as surface nanocrystalline [76], which usually occurs in both FMs and austenitic steels. However, this type of strain-induced phase transformation is not a necessary condition for the grain refinement. Furthermore, it has also been reported that the shot peening effectively cold-worked the alloy surface, producing high-density dislocations that act as rapid diffusion paths for metal, which facilitated the formation of a protective chromia scale [77]. Like grain boundaries, dislocation can also serve as rapid diffusion paths to improve the corrosion resistance of metal. However, another study revealed that the cold-working effect was not uniform, resulting in occasional nodules nucleation on the surface of the protective chromia scale [66].

However, the results from FMs Grade 91 exposed to 550 °C/1 bar CO_2_ showed that a polished surface was more resistant to oxidation than the one treated by 600-grit sandpaper (Figure 11e,f) [75]. It is believed that the potential effects of the surface finish will be a competition between the fast diffusion paths created by subsurface structural defects and the resulting surface roughness that can act as stress concentrators during oxide growth. The rougher the surface is, the greater the local stress for the growth of the scale gets. When the stress is too high, local spallation will occur, which will cause the deficiency of local Cr and lead to the formation of Fe-rich oxide nodules, thus accelerating oxidation and carburization. In conclusion, surface treatment can improve the grain boundary density or dislocation density of the alloy subsurface, which makes the alloy more resistant to oxidation. On the contrary, surface treatment may also lead to the exfoliation caused by the growth stress of the oxide layer, enhancing the oxidation of the alloy. Therefore, the surface condition has both advantages and disadvantages to the corrosion resistance of alloys.

### 2.7. Effects of Gaseous Impurities

The S-CO_2_ Brayton cycle system can be either an open or closed system. Due to leakage or degassing, air may enter the closed Brayton cycle system. Meanwhile, CO_2_ in the open Brayton cycle system comes from the oxyfuel combustion of coal, which can produce other gases. Hence, both types of systems have impurities that may greatly alter the corrosion behaviors of Fe-base and Ni-base structural alloys. It is very important to determine whether an impurity can compromise the formation of a protective scale under various conditions for the S-CO_2_ Brayton cycle power generation system. In particular, the main impurities, such as water (H_2_O), oxygen (O_2_) and sulfur dioxide (SO_2_), have a remarkable effect on oxidation resistance.

At present, much work has been done on the oxidation behavior of an alloy in wet CO_2_ (CO_2_ and H_2_O) and dry CO_2_ environments. Studies have shown that the presence of 20%H_2_O seemed to slightly reduce the oxidation rates of Fe9Cr and Fe20Cr20Ni model alloys in 650 °C/1 bar Ar-20%CO_2_ gas [66]. However, when the reaction temperature reached 818 °C, the oxidation rate of Fe9Cr model alloy and the breakaway corrosion of Fe20Cr20Ni alloy were accelerated by adding 20%H_2_O into 1bar Ar-20%CO_2_ gas [74]. It seems that the temperature can significantly affect the role of H_2_O in the oxidation of Fe-base alloys. In addition, 20%H_2_O could also slow down the corrosion rate of Ni–Cr binary alloy with low Cr contents (Ni5Cr and Ni10Cr) in 1 bar Ar-20%CO_2_ gas within the temperature range of 650 to 800 °C [78,79]. Other studies have demonstrated that the presence of 10%H_2_O had a minimal effect on the corrosion of commercial nickel-base alloys in 650–750 °C/1 bar CO_2_ environment [12,69]. In addition, it has recently been reported that the addition of 100 ppm H_2_O accelerated the oxidation rates of nickel-base alloy 740 and austenitic steel SS310 in 600 °C/300 bar S-CO_2_ environment [80]. These results indicate that the impact of impurity H_2_O on corrosion is closely related to the alloy type as well as the temperature and pressure of CO_2_.

The impurity H_2_O has significant effects on the microstructure of oxide formed on an alloy in high temperature CO_2_. The blade-shaped whiskers of Cr_2_O_3_ were observed on the top of the solid Cr_2_O_3_ layer formed on Fe20Cr0.5Si model alloy, and fine Cr_2_O_3_ grains were also observed between blade-shaped whiskers (Figure 12) [81]. It was also reported that the Ni–Cr model alloy could form similar whiskers in 650 °C/1 bar wet CO_2_ gas [78], which was composed of a loosely connected NiO scale because the hydrogen-containing species had changed the diffusion properties and chemical stability of the chromia grain boundary, resulting in the formation of the blade-shaped whiskers of Cr_2_O_3_ and fine grains between blade-shaped whiskers [82]. As shown in Figure 12d, the blade-shaped whiskers were discovered to be bicrystals, which contained an axial interface. The fast growth of blade-shaped whiskers is due to the internal interface as a quick path of diffusion [81]. Moreover, the results indicated that the surface conditions of materials also affected the formation of blade-shaped whiskers. Due to the different growth directions of whiskers, they obstruct each other during the growth, leading to large local internal stress in the scale, which may cause the local Cr_2_O_3_ scale to fail or spall [83]. The process will be repeated several times until Cr is depleted, leading to fast oxidation. However, it was observed that the Fe22Cr model alloy could maintain a protective Cr_2_O_3_ scale after 96 h exposure in 650 °C/1 bar CO_2_-30%H_2_O gas and no whiskers were observed [84]. These contradictory results cannot be explained now. In addition, it has been reported that 20%H_2_O could accelerate the scaling rate of Ni30Cr binary model alloys in 800 °C/1 bar Ar-20%CO_2_ gas [79]. It was also observed that the chromia grains formed in Ar-20%CO_2_-20%H_2_O gas (Figure 12c) were much smaller than those formed in Ar-20%CO_2_ gas (Figure 12a). This may be due to the fact that H_2_O molecules or some derivatives adsorbed on the grain boundaries of oxides inhibit the movement of grain boundaries [85,86], resulting in the formation of fine oxide grains. An increase in the number of grain boundaries may be the reason for the increase in the scaling rate of Ni30Cr binary alloy. More importantly, H_2_O seemed to cause cracks or pores at the IOZ/alloy interface of Fe9Cr model alloy in 650 °C/1 bar Ar-20%CO_2_-20%H_2_O gas [66]. The formation of these holes may be due to volume expansion caused by the oxidation of carbides or the production of methane [66]. It was also reported that H_2_O promoted the formation of cavities during oxidation in a high-temperature H_2_O-containing atmosphere [87].

The effect of impurity O_2_ on the oxidation of alloys in high temperature CO_2_ has been rarely reported. Limited works suggested that O_2_ could either accelerate or slow down the oxidation of an alloy. In Figure 13a–c, the presence of O_2_ could facilitate the formation of protective Cr_2_O_3_ scale in Ar-50%CO_2_ gas. As shown in Figure 13d, the addition of 1%O_2_ in Ar-50%CO_2_ had a small effect on the mass change at 550 °C, while the addition of 3%O_2_ in the same condition significantly reduced the oxidation rates [21]. Another study also reported that the addition of 100 ppm O_2_ could improve the stability of the chromia scale and impeded nodules formation on nickel-base alloy 740 and austenitic steel SS310 in 600 °C/300 bar S-CO_2_ environment [80]. The above results indicate that the impurity O_2_ seems to promote the formation of a protective scale. However, some studies have shown that the addition of 10 or 100 ppm O_2_ in S-CO_2_ increased the oxidation rate and facilitated the formation of nodules on the surface of nickel-base alloy Haynes 230 at 750 °C/200 bar and nickel-base alloy 625 at 750 °C/200 bar [88,89]. This may be because the quick outward diffusion of Cr caused by impurity O_2_ results in a large number of cavities underneath the chromia scale, which would induce spallation.

When both H_2_O and O_2_ were present in a high temperature CO_2_ environment, the Cr_2_O_3_ scale would volatilize [70,90,91]. The volatilization of Cr led to a serious depletion of Cr. Subsequently, Fe-rich oxide nodules formed on the surface. Another study found that a small amount of O_2_ added to wet CO_2_ gas resulted in higher carburization depth at 550 °C/1 bar on low alloy steel and 9-12Cr FMs. However, when the pressure was up to 80 bars, the situation reversed [72]. This phenomenon was not fully understood. In summary, the combination of impurity H_2_O and O_2_ is very detrimental to the resistance to oxidation of an alloy.

The addition of SO_2_ also induced the formation of whiskers on the surface of the alloy, as shown in Figure 14a,c [73]. It has been proved that the presence of whiskers caused local internal stress, leading to spallation. In addition, the impurity SO_2_ gas, which is similar to CO_2_ gas, can diffuse through the chromia scale. Then, very high values of $P_{S_{2}}$ are possible from the equilibrium established by Reaction (3) [24].
(3)$${\mathrm{SO}}_{2}\;(g)=\frac{1}{2}S_{2}\;(g)+O_{2}\;(g)$$

Therefore, sulfides were formed at the oxide layer and along the grain boundaries of the inner alloy in Figure 14b and d. The Cr-rich sulfides at the grain boundaries promote the outward diffusion of Cr and increase the scaling rate, leading to the rapid breakaway corrosion [73]. The above result shows that the Cr-rich sulfides produced by impurity SO_2_ appear to be detrimental to the oxidation of the alloy. There were other reports suggesting that adding 1%O_2_ into 550 °C/1 bar Ar-50%CO_2_-0.5%SO_2_ gas could prevent the formation of sulfides in P92 steel [92]. This is because the addition of O_2_ reduces the activity of sulfur. Furthermore, another work reported that nickel-base alloys were more resistant to vulcanization than iron-base alloys [24,27].

For Si-containing alloy, there is a competition between segregations of S and Si on the alloy surface. As shown in Figure 15, there is significant S segregated on the surface of Fe9Cr0.5Si alloy in the temperature range of 550 to 900 °C. With the increase in temperature, the segregation of S increases while the segregation of Si decreases. This indicates that Si and S repel each other at the surface, and the repulsion is strong at high temperatures. Hence, the Fe9Cr0.5Si alloy cannot form a silica layer as an additional diffusion barrier in 811 °C/1 bar Ar-20%CO_2_-20%H_2_O-0.5%SO_2_ gas [93]. Without a silica layer, the formation of the chromia scale on the surface of Fe9Cr0.5Si alloy is not stable in this condition, which will lead to quick breakaway corrosion. It has also been reported that the presence of SO_2_ gas caused spallation for commercial steels containing high levels of Si [24]. Moreover, it has been shown that impurity SO_2_ was more corrosive at a lower temperature [94]. This suggests that the effect of impurity SO_2_ on oxidation is sensitive to changes in exposure conditions or alloy composition.

The impurity SO_2_ accelerates the oxidation rate due to the formation of Cr-rich sulfides at the grain boundary or inhibition of the formation of silica layer. However, the presence of SO_2_ can partially promote the formation of the chromia layer on Fe9Cr and Fe9Cr0.5Si model alloys in wet CO_2_ (Figure 16). This is a combined effect of impurity SO_2_ and impurity H_2_O. Additionally, in complex CO_2_ environments where both O_2_ and H_2_O exist, the presence of impurity SO_2_ could accelerate the occurrence of breakaway corrosion in commercial chromia-forming steels (347H, 304H, 310S, E-Brite) [24]. However, under the same condition, it reduced the scaling rate of IOZ and impeded the Fe-rich oxide spallation on FMs Grade 91 [24]. This may be because the strong adsorption of SO_2_ at the grain boundary seems to impede the diffusion of oxygen and cause low growth stress. This may also imply that impurity SO_2_ seems more inclined to destroy the chromia layer but not the Fe-rich oxide layer in this complicated environment.

The above results indicate that caution needs to be taken when considering conventional alloys for use in application where impurities are present in a CO_2_-rich environment. In addition, the effect of impurities on oxidation is particularly sensitive to exposure conditions, which emphasizes the need to evaluate the candidate alloys within the range of expected temperatures and pressures for a given application. More research is needed to clarify the detailed process of oxidation involving gaseous impurities and the effect of other gaseous impurities on oxidation.

## 3. Carburization

### 3.1. Driving Force of Carburization

Carburization refers to the carbon deposition at the interface between the alloy substrate and oxide layer or the formation of carbides near the surface of alloy matrix under high temperature CO_2_ environment. The value of carbon activity ($a_{c}$) is quite low in high temperature CO_2_ gas. At an atmospheric pressure, the Ar-20%CO_2_ mixed gas equilibrated at 800 °C corresponds to $P_{O_{2}}$ = 1.6 × 10^−7^ atm and $a_{c}$ = 6.9 × 10^−14^. The oxygen partial pressure makes oxidation inevitable, but such low carbon activity values do not seem to pose a threat to conventional materials in terms of carburization. However, an abundance of results shows that FMs, austenitic steels and nickel-base alloys can be attacked by carbon in CO_2_ gas [29,33,37,38,39,54,96,97,98].

Why can such a low carbon activity value cause carburization? The controlling reactions are as follows:(4)$${\mathrm{CO}}_{2}\;(g)=\mathrm{CO}\;(g)+\frac{1}{2}O_{2}\;(g)$$
2CO (g) = CO_2_ (g) + C (s)(5)
(6)$$M_{X}O_{Y}\;(s)=\mathrm{XM}\;(s)+\frac{Y}{2}O_{2}\;(g)$$
and it is assumed that the whole reaction system reaches local equilibrium. The oxygen partial pressure [$P_{O_{2}}$ (4)] in Reaction (4) represents the oxygen partial pressure at oxide/gas interface, and the oxygen partial pressure [$P_{O_{2}}$ (6)] in Reaction (6) represents the dissociation pressure of the metallic oxide at oxide/metal interface. Hence, the oxygen activity of the whole system is between [$P_{O_{2}}$ (4)] and [$P_{O_{2}}$ (6)]. $P_{CO}$/$P_{{CO}_{2}}$ ratio is inversely proportional to the square root of oxygen partial pressure [$P_{O_{2}}$]. The equilibrium expression is:(7)$$\frac{P_{CO}}{P_{{CO}_{2}}}\;=\frac{K_{4}}{{\left[P_{O_{2}}\right]}^{\frac{1}{2}}}$$
where $K_{4}$ is the equilibrium constant for Reaction (4). [$P_{O_{2}}$ (4)] is much larger than [$P_{O_{2}}$ (6)]. Therefore, *b* = $P_{CO}$/$P_{{CO}_{2}}$ at oxide/metal interface is much greater than that at oxide/gas interface. At last, the carbon activity ($a_{c}$) is calculated from the Boudouard [44] equilibrium Reaction (5) as:(8)$$a_{c}=k_{5}\frac{P_{c0}^{2}}{Pco_{2}}$$
(9)$$a_{c}=k_{5}\frac{b^{2}P_{T}}{1+b}$$
where $K_{5}$ is the equilibrium constant for Reaction (5). The Equation (9) is derived from Equation (8). The total external pressure, $P_{T}$, is equal to $P_{{CO}_{2}}$ plus $P_{CO}$, which ignores the very small oxygen partial pressure [33]. According to Equation (9), when the total external pressure, $P_{T}$, increases, the carbon activity will also increase, thus increasing tendency to carburization. It also indicates that the carbon activity at the oxides/metal interface is considerable as the local *b* is high, which is several orders of magnitude greater than that in CO_2_ atmosphere. As a result, it is possible for carbon to accumulate at the interface between oxides and metal.

The above analysis has explained the driving force of carburization in high temperature CO_2_ environment. However, studies [99] have shown that carbon was insoluble in oxides and the lattice diffusion was not possible. According to the previous research results, carbon reached the interface between the metal and the oxide in the form of molecules, such as CO or CO_2_, through grain boundaries [100], microchannels and nanopores [99] in the oxide layer. However, there was no direct evidence for this conjecture. Recently, as shown in Figure 17, David J. Young et al. [101,102,103] analyzed the Cr_2_O_3_ layer on the surface of Fe20Cr model alloy by atomic probe and found that carbon was enriched in the sub-monolayer level of grain boundary, which was supposed to be in the form of molecular oxide. In this form, moving is possible, and the penetration through oxide scale leads to carburization of the underlying alloy. In this experiment, it was demonstrated that CO or CO_2_ penetrated to the interface of oxide/metal through the grain boundaries of oxide.

### 3.2. Consequences of Carburization

When CO and CO_2_ cross the grain boundaries to reach the oxide/substrate interface, carbon will accumulate at this interface, as mentioned in the driving force of carburization. At present, there are two main forms of accumulated carbon layer: amorphous carbon layer and graphite. As shown in Figure 18b, the formation of amorphous carbon layer was observed at the oxide/matrix interface of Ni14Cr binary alloy after 200 h corrosion in 600 °C/20 MPa S-CO_2_ [97]. The graphite was observed in Fe9Cr1Mo steel (Figure 18c), which was corroded in 600 °C/42 bar CO_2_ gas for 20,000 h [104]. The precipitation of graphite due to carbon saturation was supported by the modeling [104]. As for the formation of amorphous carbon layer, no good explanation has been provided so far. Furthermore, carbon sometimes accumulates in holes in the oxide layer, which will increase the probability of spallation and accelerate breakaway corrosion [54].

As shown in Figure 18a, martensite was observed in Fe9Cr alloy after exposure to 800 °C/1 bar Ar-20%CO_2_ for 20 h, which was a single ferritic phase before the reaction [33]. This is because the graphite or amorphous carbon layer can dissolve in ferritic alloys and transform ferrite into austenite at high temperature, which will transform into martensite at room temperature [33,105]. Moreover, it can also diffuse into the internal metal to form carbides, such as Cr_3_C, Cr_7_C_3_ and Cr_23_C_6_, which pins the movement of Cr in chrome-depletion zone and deteriorates the corrosion resistance of the alloy [106]. It has been reported that the kinds of Cr-rich carbides mainly depended on carbon activity, Cr content and temperature [79,107]. In addition, other metallic carbides, such as Nb-rich, Mo-rich, V-rich, Fe-rich and W-rich carbides, may also form in internal alloy.

Another risk is that carburization or carbon deposition can cause ‘metal dusting’ [108]. Metal dusting is a catastrophic carburization phenomenon, which occurs in very high carbon activity environment. The carburization behavior of 304 stainless steel was investigated during thermal cycling in 700 °C/1 bar CO/CO_2_ and 680 °C/1 bar CO/H_2_/H_2_O [109]. The CO/CO_2_ gas with $a_{c}$ = 7 caused surface spallation that was related to thermal cycling but no metal dusting after 528 cycles. For the CO/H_2_/H_2_O gas with $a_{c}$ = 19, graphite could be quickly formed, and metal dusting could be produced. Discrete oxide precipitates and a deep local surface dent suggested the occurrence of metal dusting (Figure 19a,b). Moreover, the Murakami’s reagent revealed that there were many fine intragranular and intergranular carbides present in the substrate of an alloy (Figure 19c,d). It has been reported that internal carbides were oxidized in situ to generate spinel and carbon, resulting in volume expansion, which fragmented the metal and produced metal dusting [110,111]. The occurrence of metal dusting will cause the surface of an alloy to decompose into a powder mixture of carbon, metals, oxides and carbides. Carburization occurs in most alloys, and it is necessary to find an effective way to mitigate it by preventing the penetration of carbon.

### 3.3. Effects of Environmental Parameters on Carburization

Like oxidation, carburization is also influenced by environmental factors. As shown in Figure 20, the increasing pressure of CO_2_ would increase the carbon mass gain of FMs T91 [17]. Meanwhile, it has also been reported that the depth of carburization and micro hardness at oxide/matrix interface in FMs F91 increased with increasing CO_2_ pressure [48]. Another study also showed that graphite precipitation at oxide/matrix interface would be accelerated in FMs CrMoV due to increase in CO_2_ pressure [41]. These results suggest that the increase in gas pressure will elevate the carbon activity and enhance the rate of carburization, which is consistent with that mentioned in Section 3.1 (Equation (9)). In addition, the higher the temperature, the deeper the carburization (Figure 21) [40]. This may be because the increase in temperature accelerates the diffusion of carbon, which leads to an increase in the depth of carburization. More importantly, compared to the flowing gas, the static gas not only reduced the oxidation rate but also decreased the carburization rate (Figure 9 and Figure 20). Moreover, it was observed that the carburization rate decreased with decreasing gas flowing rate in FMs T91 (the mc = 5.2 g/m^2^ in 250 bar flowing CO_2_ and the mc = 3.9 g/m^2^ in 250 bar static CO_2_). This is because carbon in the static atmosphere is consumed without replenishment, resulting in a decrease in carbon activity, which leads to the reduction in carburization.

The effects of pressure and temperature on the carburizing rate of austenitic steels and nickel-base alloys have also been reported. It has been shown that an increase in pressure increased the thickness of the amorphous carbon layer in austenitic steel 800HT, nickel-base alloy 600 and nickel-base alloy 690 after exposure to 600 °C/(0.1–20 MPa) CO_2_ after 1000 h, but it had no effects on the depth of carbides formation [65]. This may be due to the enhancement of short-circuit diffusion of CO_2_ or CO under high pressure, which accelerates carburization [65,112,113]. Meanwhile, it has also been reported that, when the temperature increased (550–650 °C), the depth of carbides formation in austenitic steel 800HT increased, while the carburizing rate of nickel-base alloys 600 and 690 seemed to be barely influenced. Overall, the increase in pressure mainly increases the thickness of the amorphous carbon layer or accelerates the precipitation of graphite, while the increase in temperature mainly increases the diffusion rate of C, forming coarser and deeper carbides in the matrix.

### 3.4. Carburization Resistance

The different oxide layers show different resistances to carburization. For FMs Fe20Cr, below the Fe-rich oxide and Fe–Cr spinel oxide layer, many carbides were formed intragranularly and intergranularly (Figure 22a). Meanwhile, small amounts of intergranular carbides could be observed under the Cr_2_O_3_ layer (Figure 22a). However, no graphite was observed in Figure 22a. This may be due to the short reaction time (120 h) or low pressure (1 bar). Although the chromia scale cannot completely suppress the carburization, it has stronger resistance to carburization than the Fe-rich oxide layer or Fe–Cr spinel oxide layer. In addition, the alumina, which is also a very dense and protective scale, is more resistant to carburization than Cr_2_O_3_ scale [57,68,96]. There was no amorphous carbon layer, graphite or carbides formed under the alumina scale in ODS Fe–Cr–Al alloy (Figure 22b). This is because carbon does not segregate at the grain boundaries of alumina, which is different from the case with chromia scale [68]. Nevertheless, the free carbon films were also observed on the surface of alumina scale, which had little effect on the corrosion performance of the alloy. Such free carbon film has also been observed on other alloys [25,46], which may be caused by decomposition of CO_2_ or the Boudouard reaction [57]. In conclusion, the order of decreasing carburization resistance is alumina, chromia and Fe-rich oxide.

The resistance to carburization is not only related to the oxide layer but also related to the type of alloy. Many carbides and the amorphous carbon layer were observed in austenitic steel 800HT, but only the amorphous carbon layer was present in nickel-base alloy 690 (Figure 22c and d). Compared with iron-base alloys, it may be due to the low activity of Cr in nickel-base alloy and the low solubility of Ni in carbides hindering the formation of internal carbides [29,114]. In addition, it has also been shown that the diffusivity of C decreased with the increase of Cr and Ni content [98,115,116]. These previous results indicate that chromia-forming nickel-base alloys have higher carbonization resistance than chromia-forming iron-base alloys. Surprisingly, the intergranular carbides were revealed by etching in Ni25Cr and Ni30Cr model alloys after exposure to 650 °C/1 bar Ar-20%CO_2_ gas for 500 h but not in Ni20Cr [47]. When the temperature was up to 800 °C, the Ni20Cr model alloy also formed the intergranular carbides. Hence, it is thought that temperature and Cr content are the key factors for carbides formation in nickel-base alloy.

In conclusion, alumina film and nickel-base substrate have better carburizing resistance. Perhaps, Ni–Cr–Al alloys can avoid the carburization in high temperature CO_2_. Recently, a double-layer barrier to prevent the carburizing of alloys was proposed. It was suggested that alloying Si or Mn could form an additional diffusion barrier layer (SiO_2_ or MnCr_2_O_4_) to resist carburization [117,118,119,120]. Additionally, it has also been found that adding SO_2_ or H_2_O or both in CO_2_ could reduce carburization to some extent [51,73,74,80,93,95,118,121]. This may be due to the higher adsorption tendency of SO_2_ or H_2_O compared with CO_2_ in grain boundaries of oxide [122,123]. However, other reports suggested that impurity H_2_O would accelerate carburization in Ni30Cr2Ti model alloy [62] and austenitic steel SS310 [81]. This unexpected effect is attributed to the change in grain boundary characteristics of chromia caused by water vapor absorption. Further efforts should be made to mitigate the carburization of structural alloys in the S-CO_2_ Brayton cycle.

## 4. Stress Corrosion

### 4.1. Effects of Corrosion on Mechanical Behavior

In a real S-CO_2_ Brayton cycle system, structural alloys are subject to both stress and corrosion. The exposure to S-CO_2_ will cause oxidation and carburization of the alloy. It has been reported that the depth of carburization was greater than the depth of oxidation [104]. Therefore, carburization may have a greater impact on mechanical behavior than oxidation. Under stress, the oxides or carbides will act as potential crack initiation sites, especially intergranular oxides or intergranular carbides, which will cause the premature failure of materials. Hence, investigating the effects of corrosion on the mechanical behaviors of materials or studying the stress corrosion behaviors of materials is of great significance to the reliability of the S-CO_2_ Brayton cycle system.

The mechanical behaviors of FMs Grade 92 exposed to (450 °C, 550 °C)/20 MPa S-CO_2_ for 1000 h were investigated [20]. The results of tensile performance, given in Table 1, demonstrated that an increase in strength with decrease in elongation occurred in both base and welded materials after S-CO_2_ exposure. More importantly, this strengthening effect increased with increasing temperature in the S-CO_2_ environment. In contrast, the mechanical properties of FMs Grade 92 hardly changed after aging under the same condition without CO_2_. This might be because the laves-phase precipitate was not dense enough to affect the mechanical property [124]. It has been reported that the exposure to S-CO_2_ would carburize the alloy, and the carburization rate and the formation of carbides increased with temperature [40]. This explains why the strengthening effect increases with increasing temperature. Furthermore, some studies [29,65,125,126] have reported that the mechanical performance of base materials or welded samples or diffusion-bonded joints of austenitic steel 316SS and 800HT were greatly compromised after exposure to high temperature S-CO_2_. The fracture mode gradually changed from transgranular fracture to intergranular fracture at room temperature due to the carburization in high temperature S-CO_2_.

The effect of the thickness of the specimen on the degradation of the mechanical properties of FMs P91 under a direct-fired CO_2_ environment was also studied [34]. A significant reduction in elongation was observed in 2.54 mm thick FMs P91 exposed to 650 °C/1 bar CO_2_ (Figure 23). The thinner samples (0.5 mm thick FMs P91), which were exposed to 650 °C/1 bar CO_2_, changed from ductile fracture to brittle fracture (Figure 23). From these results, it is discovered that the ratio of carburizing depth to original thickness of FMs P91 can significantly affect the degree of degradation in a mechanical property. This implies that alloy thickness is a key parameter as well when used in a S-CO_2_ Brayton cycle system.

According to previous results, the carburization resistance of nickel-base alloys is generally higher than that of iron-base alloys, especially 9-12Cr FMs and austenitic steels. Therefore, the mechanical properties of nickel-base alloys may be less affected by carburization in the CO_2_ environment. The experimental results [29] showed that the ultimate tensile strength and elongation of nickel-base alloys (600, 690) underwent no remarkable changes compared with the as-received samples for either S-CO_2_ exposed samples or bulk-aged samples (Figure 24). As for the austenitic steel 800HT, the elongation decreased with increasing temperature in S-CO_2_ (Figure 24). This is because nickel-base alloys form only amorphous carbon layers at the oxide/substrate interface, while austenitic steels form a large number of internal carbides in addition to amorphous carbon layers (Figure 24c). Moreover, it was also reported that the base material (nickel-base alloy 600) and the fusion weld (nickel-base alloy 182) could maintain their mechanical performance after exposure to (550 °C, 650 °C)/20 MPa CO_2_ for 1000 h [127]. However, another study showed that the fracture occurred along the bond line in brittle mode for the diffusion-bonded joints of nickel-base alloy 690 exposed to 650 °C/20 MPa S-CO_2_ [128]. The reduction in strength and elongation after exposure to 650 °C/20 MPa S-CO_2_ was due to the coarsening of the carbides along the bond line. In summary, compared with austenitic steels and 9-12FMs, the mechanical behaviors of nickel-base alloys are least affected by the high temperature CO_2_ corrosion.

### 4.2. Stress Corrosion Cracking

These experiments analyzed above could not represent the actual situation during plant operation because the tensile samples did not bear the load during S-CO_2_ corrosion. Recently, stress corrosion cracking of alloys in S-CO_2_ has also been investigated limitedly. The author used U-bend samples to evaluate the susceptibility to stress corrosion cracking of austenitic steel 316 and nickel-base alloy Haynes 230 in 650 °C/20 MPa S-CO_2_ [3]. The results showed that the applied stress did not seem to induce the stress corrosion cracking of the alloy. Additionally, the creep tests were carried out for nickel-base alloy 600 in air, CO_2_ and S-CO_2_ (20 MPa) at 650 °C [129]. At the low applied stress of 135 MPa, the creep rupture life in air or CO_2_ was about three times longer than that in S-CO_2_ (Table 2). At the high applied stress of 160 MPa, the creep rupture life in air or CO_2_ was approximately twice longer than that in S-CO_2_ (Table 2). It was clear that the influence of the test environment on creep rupture was more remarkable at low applied stress. Moreover, with the increase in stress, the creep rupture life decreased gradually. As shown in Figure 25a, the inside of the crack was filled with Ni-rich oxide and Fe-rich oxide, while Cr-rich oxide was mainly present in the flanks of the crack and crack tip. The process might involve the cracking or spallation of Cr-rich oxide by applied stress, then Fe-rich oxide and Ni-rich oxide filled the inside of the crack. Besides, Al-rich oxide and Ti-rich oxide were observed in the crack tip region. The grains near the crack tip had different orientations, indicating that the crack propagated along the grain boundary aided by grain boundary oxidation ahead of the crack tip (Figure 25b). This is because these grain boundary oxides are brittle and prone to crack, which accelerates crack propagation and leads to premature fracture. Therefore, it is believed that the cracking or spalling of Cr-rich oxide on the surface of the alloy leads to the initiation of cracks; subsequently, the crack propagates along the grain boundary, which was firstly oxidized, thus accelerating the rupture of nickel-base alloy 600. In addition, the creep of MARBN-type 9Cr martensitic steel in 650 °C/1 bar CO_2_ atmosphere was also studied [130]. The results demonstrated that the carburization in sub-surface zone facilitated the propagation of cracks initiated in the oxide scale.

At present, there are very limited reports about the stress corrosion cracking of alloys or about the mechanical behaviors of alloys after exposure to high temperature CO_2_. However, from the few reports available, it is found that carbides formation greatly reduces the mechanical properties of alloys, leading to a gradual transition from transgranular fracture to intergranular fracture. In addition, the stress corrosion cracking of nickel-base alloy 600 in S-CO_2_ indicates that the crack propagates along the intergranular oxide, and the stress corrosion cracking of MARBN-type 9Cr martensitic steel in a CO_2_ environment demonstrates that carburization can also promote crack propagation. The above evidence indicates that the oxidation or carbonization of grain boundaries will greatly reduce the strength of grain boundaries. In other words, in the presence of stress, the failure of the alloy will be greatly accelerated in an intergranular mode under a CO_2_ environment. Therefore, the grain boundary corrosion behavior of the alloy is well worthy of in-depth study, which is closely related to stress corrosion cracking in the S-CO_2_ Brayton cycle system.

## 5. Conclusive Remarks

Although the corrosion behavior of alloys in high temperature CO_2_ is more complex than the conventional gaseous oxidation, the basic principle is almost the same. Given that the corrosion kinetic is still controlled by diffusion in high temperature CO_2_, it can be evaluated by the parabolic rate constant $K_{w}$ or $K_{P}$. The corrosion kinetic of an alloy is closely related to the corrosion products. In general, a protective chromia scale formed on nickel-base alloys and austenitic steel, while an Fe-rich oxide layer formed on 9-12Cr FMs. This makes the corrosion kinetic of austenitic steel as low as that of nickel-base alloy. However, austenitic alloy is more prone to breakaway corrosion, which will cause quick oxidation and carburization. Therefore, nickel-base alloys have better oxidation resistance than austenitic steels and 9-12Cr FMs. The oxidation behavior of an alloy is greatly affected by environmental parameters (temperature, pressure or gas flowing rate), surface treatment (polishing or sanding) and gaseous impurity (H_2_O, O_2_ or SO_2_). For environmental parameters, temperature and pressure have both advantages and disadvantages to the corrosion resistance of alloys. Moreover, the static CO_2_ environment can reduce the oxidation rate and carburization rate of the alloy compared with the flowing CO_2_ condition. For surface treatment, the potential effects of the surface finish would be a competition between the fast diffusion paths for the active element created by sub-surface structural defects and the resulting surface roughness, which can act as stress concentrators during oxide growth. If the stress is too large, it will lead to the peeling of the oxide film, which will accelerate the oxidation rate of the alloy. For gaseous impurities, their effects are extremely sensitive to exposure conditions. The impurity H_2_O or O_2_ has both advantages and disadvantages regarding the corrosion resistance of an alloy. However, when both are present together, they can greatly accelerate the oxidation of the alloy. The presence of impurity SO_2_ will accelerate the oxidation of alloys in a dry CO_2_ environment and reduce the oxidation rate in a wet CO_2_ environment. Furthermore, in a more complex environment (CO_2_–H_2_O–O_2_), the impurity SO_2_ is more aggressive at lower temperatures and more prone to damage the chromia scale.

Carburization is another major degradation mode in addition to oxidation. It normally occurs in three steps. At first, CO or CO_2_ passes through the grain boundary of oxide to the interface between the oxide and matrix. Subsequently, CO or CO_2_ accumulates at the oxide/substrate interface and forms an amorphous carbon layer or graphite or both of them. Finally, the accumulated carbon layer will diffuse into the matrix to form carbides or make ferrite transform to martensite. The ingress of excessive carbon can greatly degrade the material. For example, metal dusting is a catastrophic carburization phenomenon, by which the surface of an alloy decomposes into a powder mixture of carbon, metals, oxides and carbides. Like oxidation, the carburization rate of an alloy is also associated with environmental parameters, corrosion products and the types of alloys. As for the environmental parameters, the increase in temperature and pressure accelerates the carburizing rate of the alloy. As for the types of alloys, the carburization resistance of nickel-base alloy is better than that of iron-base alloy. As for the corrosion products, alumina can stop the carburization, chromia can slow down the carburization and hematite/magnetite will undergo quick carburization.

The load-bearing capacity during service is a key element to the structural material and should be evaluated thoroughly. The mechanical performances of 9-12Cr FMs and austenitic steel are greatly degraded after exposure to high temperature CO_2_, which is caused by severe carburization in the grain boundary. Meanwhile, the fracture modes of austenitic steel and 9-12Cr FMs change from transgranular fracture to intergranular fracture. This indicates that serious carburization in the grain boundary will reduce the strength of the grain boundary, making the alloy more prone to intergranular fracture. Moreover, the mechanical performance of nickel-base alloy (600, 690), which has better carburization resistance in high temperature CO_2_, hardly degrades after the exposure test. However, the formation of intergranular oxides in nickel-base alloy 600 will accelerate the stress corrosion cracking of alloys when stressed during an exposure test. Like carburization in the grain boundary, intergranular oxidation also degrades the mechanical performance of an alloy.

## 6. Scopes for Future Research

In order to have higher thermal efficiency, the key components will be used at a high temperature (more than 650 °C) and high pressure (more than 200 bar) in a S-CO_2_ Brayton cycle power generation system, which poses a great challenge to the integrity of the structural alloy. In this harsh environment, it is necessary to ensure that the lifetime of key components is not less than 20 years, while that for other non-removable components should be as high as 60 years. Until now, a large number of studies have focused on the corrosion behavior of conventional alloys in high temperature CO_2_. Therefore, the main candidate structural materials for a S-CO_2_ Brayton cycle system may be nickel-base alloys, austenitic steels and FMs. According to previous test results, the nickel-base alloys and alumina-forming alloys possess excellent carburization resistance and oxidation resistance in this environment. Perhaps, they will be used for fabricating key components, such as a heat exchanger in the S-CO_2_ Brayton cycle power generation system. Although extensive work has been done in this field so far, further research efforts should be made to address some unresolved issues.

According to previous results, the static CO_2_ environment can reduce the oxidation and carbonization rates of alloys. This implies that the corrosion rate of an alloy must be associated with the CO_2_ flowing rate. Moreover, the CO_2_ flowing rate is also closely related to heat transfer, which affects the thermal efficiency of a power plant. Therefore, it is very critical to choose the CO_2_ flowing rate in a S-CO_2_ Brayton cycle power generation system to balance the corrosiveness of the environment and the thermal efficiency. However, the reports on the effect of the CO_2_ flowing rate on corrosion are very limited at present. From the current reports, the CO_2_ flowing rates used in the laboratory are different and there is no standard. This makes it difficult to relate the data from experiments to the real S-CO_2_ Brayton cycle systems. Hence, it is necessary to study the effect of the CO_2_ flowing rate in a S-CO_2_ Brayton cycle system on the corrosion behavior of alloys so as to further improve the relevance of the experimental data.

The candidate materials are subject to both oxidation and carburization in the S-CO_2_ environment. Currently, it is generally observed that the depth of carburization is much greater than the depth of oxidation, which means carburization and oxidation may occur in sequence in the matrix. In addition, some experimental results show that a mixed layer is formed in internal matrix, which contains both carbides and oxides. There is no specific explanation for the formation of such mixed layer. The interaction between oxidation and carbonization in S-CO_2_ conditions is not well understood, and there is a lack of relevant research. Furthermore, in addition to carburization, conventional alloys at such high temperatures also precipitate carbides at grain boundaries due to aging. It is also unknown whether the precipitation of such carbides will further affect the corrosion behavior of alloy. Therefore, the interaction between oxidation and carbonization should be investigated to better understand the entire corrosion process of alloy during the application in S-CO_2_ environment.

The effects of impurities on the corrosion of alloys should be considered in the S-CO_2_ environment. This is because the S-CO_2_ Brayton cycle system includes an open system and closed system, both of which are expected to have some impurities. The effects of impurity H_2_O, O_2_, SO_2_ and their interaction on the corrosion behavior of alloys in CO_2_ have been widely reported at various temperatures. Studies show that the effects of these impurities on corrosion not only exhibit a synergistic mechanism but also a competitive mechanism, which is also very sensitive to exposure conditions. This warns us that it is necessary to evaluate the candidate alloys within the range of expected temperatures and pressures for a given application. However, nearly all the experiments only considered the effect of temperature on the roles of impurities, while overlooking the effect of pressure or gas flowing rate. Hence, these results can only be used temporarily as a reference for the application of S-CO_2_ Brayton cycle systems because it is not known whether the pressure or gas flowing rate also affect the role of impurity. In addition, impurities are not limited to O_2_, H_2_O and SO_2_. Due to leakage or degassing, air (N_2_, O_2_ and H_2_O) may enter the closed Brayton cycle system. The open Brayton cycle system powered by the oxyfuel combustion of coal will face other gases, such as HCl, SO_3_ or oxynitride. Hence, more extensive research is needed to improve the understanding of the impurity effect and make life prediction models for alloys exposed to these complex environments more realistic.

More importantly, research shows that the oxidation or carbonization of the grain boundary will weaken the strength of the grain boundary, causing the failure mode to gradually change from transgranular fracture to intergranular fracture. Hence, grain boundary oxidation or carbonization can be the cause of crack initiation or propagation. This implies that the corrosion behaviors of grain boundaries can be the key to understanding the mechanism of crack initiation or propagation. This is because an understanding of preferential intergranular oxidation is necessary to predict intergranular stress corrosion cracking. However, there are quite few reports on the grain boundary corrosion behavior at present and there are many unresolved issues. Firstly, the grain boundary corrosion behavior is drastically different from the intragranular corrosion behavior and should be studied through detailed microstructure analysis. Secondly, the effect of grain boundary type on grain boundary corrosion behavior should be investigated to identify the most vulnerable grain boundary type. Lastly, the effects of solute segregation and grain boundary precipitation on grain boundary corrosion behavior should be surveyed. Resolving these issues can provide valuable guidance for subsequent work, such as grain boundary engineering, the development of new materials and modeling the failure of materials.

Furthermore, in most studies regarding the degradation of a mechanical property, the changes in the mechanical properties of alloys are evaluated after exposure to high temperature S-CO_2_. These results cannot represent the actual situation during plant operation because the tensile samples do not bear the loading during S-CO_2_ corrosion. Therefore, alloys are needed to be tested for stress corrosion, such as creep, crack growth, fatigue and crack initiation tests in a high temperature S-CO_2_ environment. It is crucial to understand the crack initiation mechanism and crack propagation mechanism of an alloy, which provide the basis to evaluate the long-term performance of materials under the synergistic actions of chemical and mechanical processes in S-CO_2_ cycle power generation systems.

The current candidate materials, i.e., nickel-base alloys, austenitic steels and FMs, are not corrosion-resistant enough in the S-CO_2_ environment. From the current reports, carburization results in quick failure of the austenitic steels and FMs, and intergranular oxidation accelerates the failure of nickel-base alloys. Therefore, the candidate materials for the S-CO_2_ Brayton cycle system should not be limited to traditional alloys. Developing new materials, such as high entropy alloys, ceramics and composites, is well worth considering in the future. Moreover, coating can greatly improve the corrosion resistance of an alloy, which eliminates carburization and makes relatively cheap alloys (austenitic steels and FMs) usable at high temperatures in S-CO_2_ Brayton cycle systems. However, it is very complicated to design a functional coating for specific environmental conditions, alloy composition and component geometry. Therefore, a great deal more research is needed to extend the database for coating technology.

## Figures and Tables

**Figure 1 materials-15-01331-f001:**
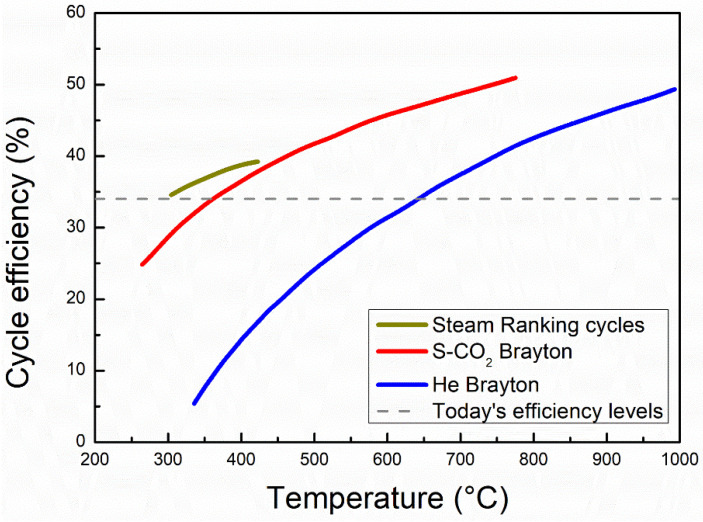
Comparison of theoretical efficiency in different systems. Data were obtained from reference [3].

**Figure 2 materials-15-01331-f002:**
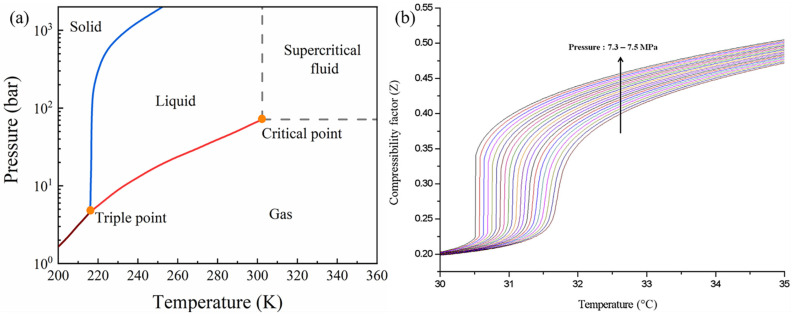
(**a**) Phase diagram of CO_2_. Data were obtained from reference [3]. (**b**) CO_2_ compressibility factor near the critical point [2]. Reproduced with permission from Jeong Ik Lee (corresponding author).

**Figure 3 materials-15-01331-f003:**
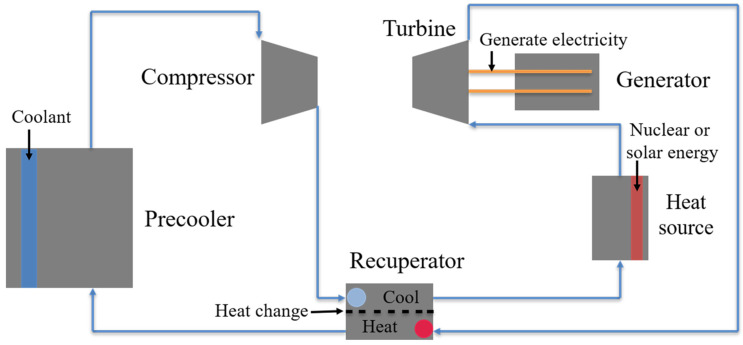
A simple schematic of the closed Brayton cycle system.

**Figure 4 materials-15-01331-f004:**
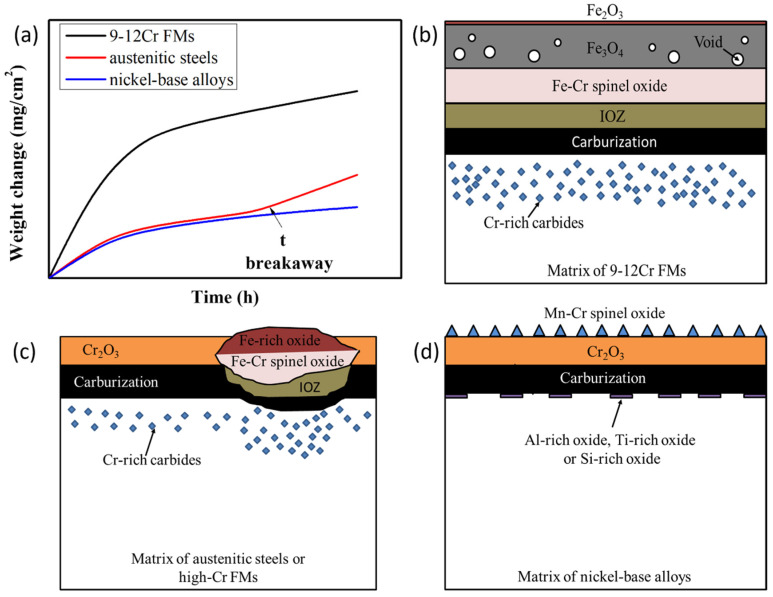
(**a**) Schematic diagram of typical corrosion kinetics of conventional alloys in high temperature CO_2_ [36,37,38,39]. Schematic diagrams of typical cross-section of conventional alloys corroded in high temperature CO_2_ after time t. (**b**) 9-12Cr FMs [17,24,34,38,40,41,42,43,44,45]. (**c**) Austenitic steels or high-Cr FMs [22,25,29,33]. (**d**) Nickel-base alloys [28,29,46,47]. The carbides above are caused by carburizing, and the precipitation of carbides after aging is not considered. The carburization (black) represents amorphous carbon layer or graphite.

**Figure 5 materials-15-01331-f005:**
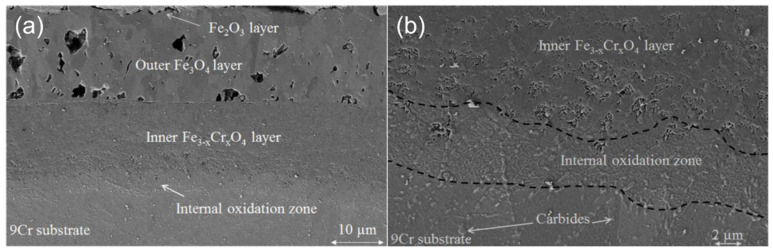
(**a**) Scanning electronic microscopy (SEM) cross-section image of the oxide layer on FMs T91 sample after corrosion test under flowing CO_2_ at 550 °C/1 bar for 350 h. (**b**) Internal oxidation zone of (**a**) [17]. Reproduced with permission from Elsevier.

**Figure 6 materials-15-01331-f006:**
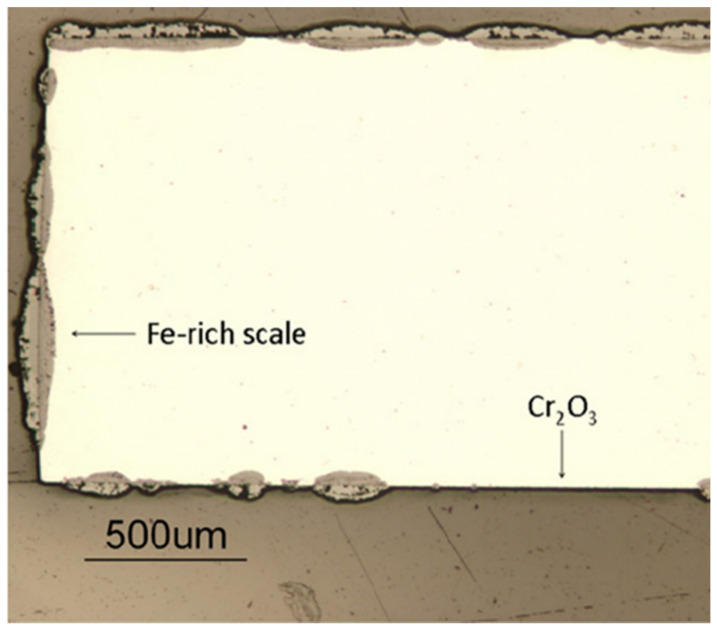
The cross-section of Fe20Cr after 120 h exposure to Ar-20%CO_2_ at 650 °C/1 bar [33]. Reproduced with permission from Elsevier.

**Figure 7 materials-15-01331-f007:**
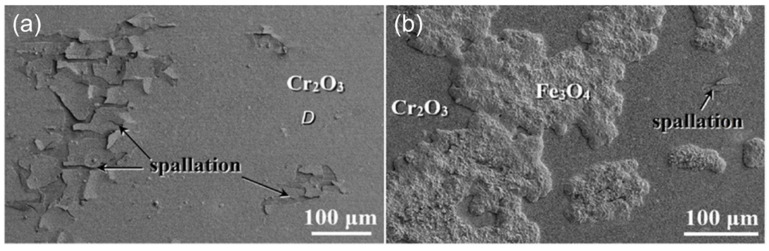
The low magnification SEM image of 347H stainless steel after exposure to S-CO_2_ at 650 °C/20 MPa for 500 h (**a**) and 1000 h (**b**) [19]. Reproduced with permission from Elsevier.

**Figure 8 materials-15-01331-f008:**
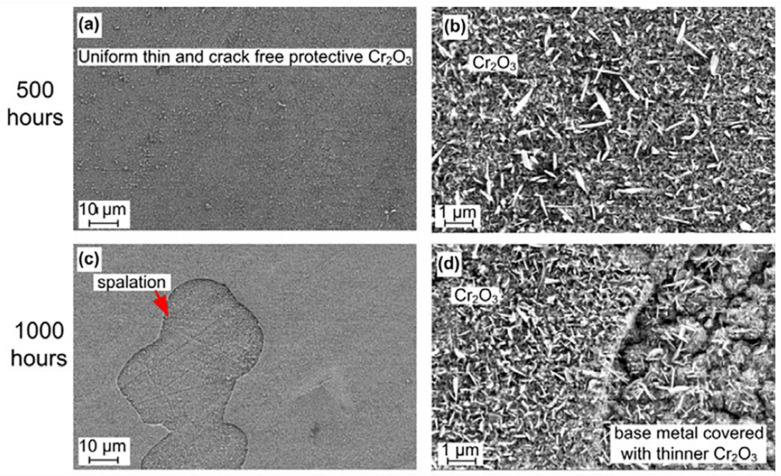
Surface oxide morphology of alloy 625 samples exposed to S-CO_2_ at 650 °C/20 MPa for (**a**,**b**) 500 h, (**c**,**d**) 1000 h [46]. Reproduced with permission from Elsevier.

**Figure 9 materials-15-01331-f009:**
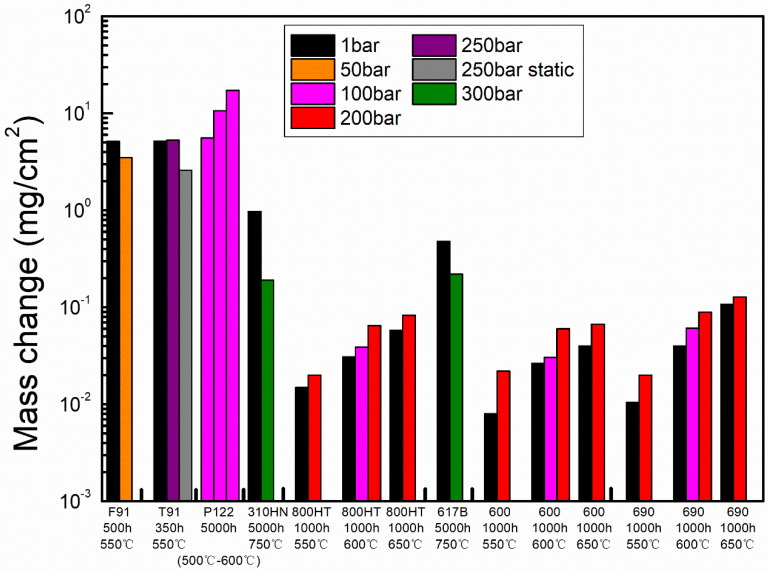
Correlation of mass change with temperature and pressure for different alloys in CO_2_ gas. Data were obtained from references [17,48,65,70].

**Figure 10 materials-15-01331-f010:**
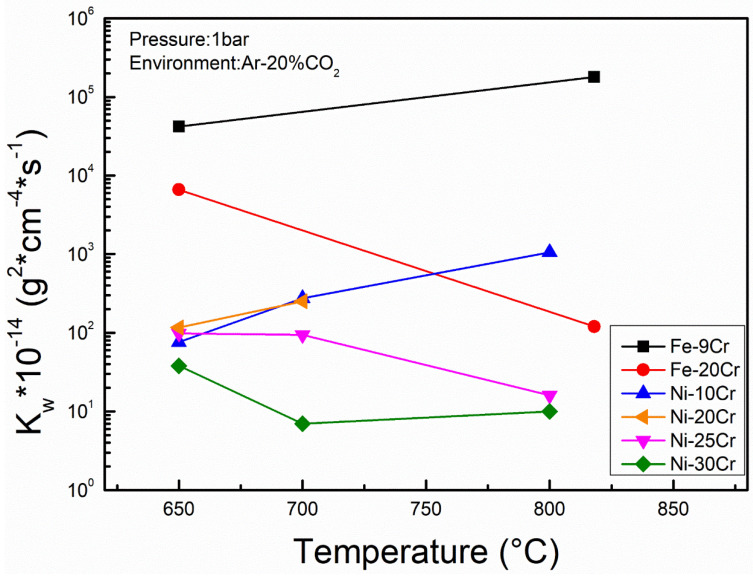
The correlation between the weight gain constant and temperature of Ni–Cr and Fe–Cr binary alloy in Ar-20%CO_2_ gas. Data were obtained from references [47,73,74].

**Figure 11 materials-15-01331-f011:**
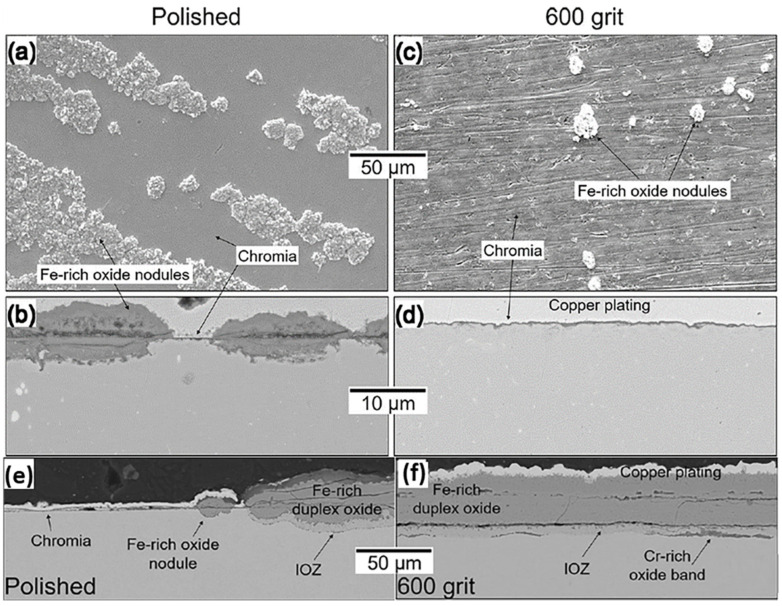
Surface and cross-sectional SEM images of austenitic steel 347H with two different surface treatments after exposure to 20 MPa/550 °C S-CO_2_ for 1500 h. (**a**,**b**) Polished; (**c**,**d**) 600-grit. Cross-sectional SEM images of FMs Grade 91 with two different surface finishes after exposure to 550 °C/1 bar CO_2_ for 1500 h. (**e**) Polished; (**f**) 600 grit [75]. Reproduced with permission from Springer Nature.

**Figure 12 materials-15-01331-f012:**
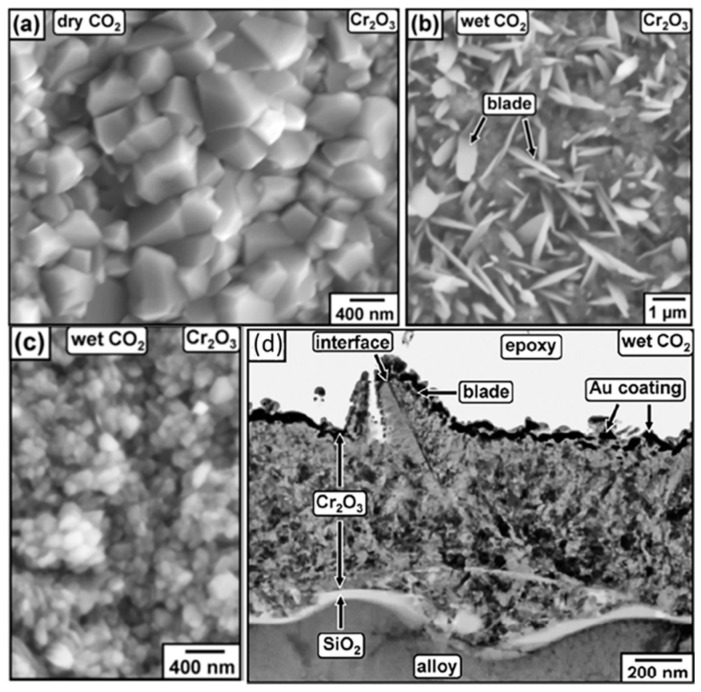
SE-SEM top views of Fe20Cr0.5Si model alloy (ground with 1200 grit paper) after reaction for 240 h in 818 °C/1 bar: (**a**) Ar-20%CO_2_ and (**b**) Ar-20%CO_2_-20%H_2_O. (**c**) High magnification image of the solid surface between blade-shaped whiskers in (**b**). (**d**) BF-TEM cross-sectional images of Fe20Cr0.5Si model alloy (ground with 1200 grit paper) after reaction for 240 h in 818 °C/1 bar Ar-20%CO_2_-20%H_2_O [81]. Reproduced with permission from Elsevier.

**Figure 13 materials-15-01331-f013:**
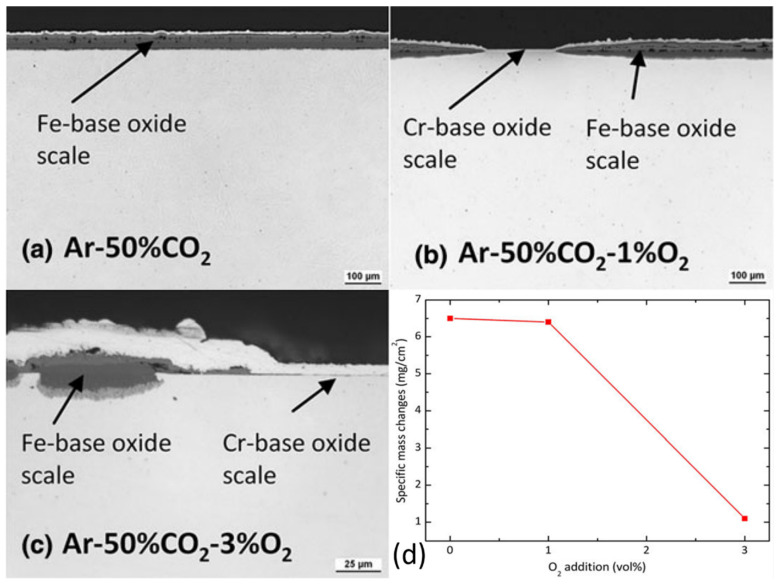
Cross-sections of P92 exposed at 550 °C in (**a**) Ar–50%CO_2_, (**b**) Ar–50%CO_2_–1%O_2_ and (**c**) Ar–50%CO_2_–3%O_2_. (**d**) Effect of excess oxygen on the oxidation mass gain of P92 after 250 h in Ar–50%CO_2_ at 550 °C [21]. Reproduced with permission from Springer Nature.

**Figure 14 materials-15-01331-f014:**
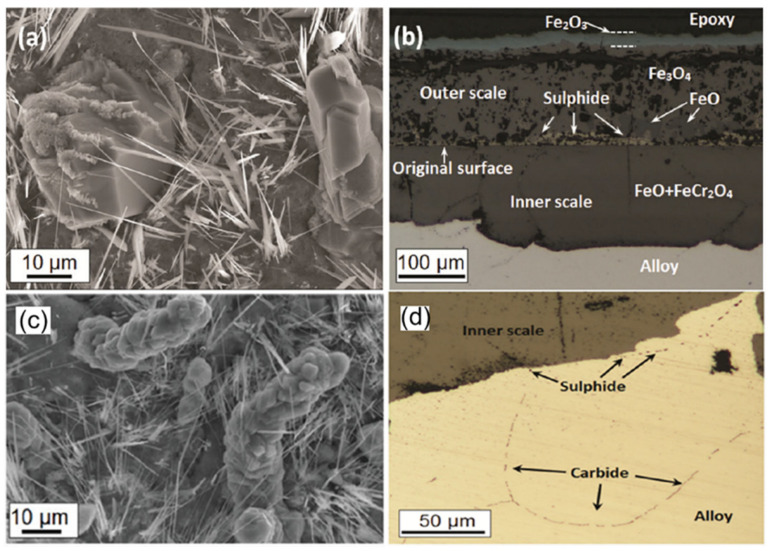
SEM surface morphology (**a**) and cross-section (**b**) images of Fe9Cr alloy exposed to Ar-20%CO_2_-0.5%SO_2_ gas for 500 h at 650 °C/1 bar. Surface morphology (**c**) and cross-section images after etching (**d**) of Fe20Cr alloy exposed to Ar-20%CO_2_-0.5%SO_2_ gas for 500 h at 650 °C/1 bar [73]. Reproduced with permission from Springer Nature.

**Figure 15 materials-15-01331-f015:**
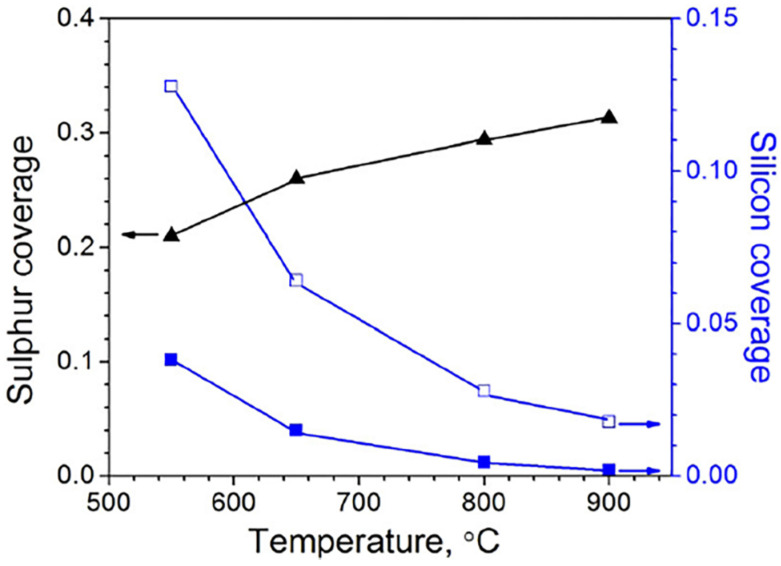
The equilibrium coverage of sulphur and silicon during their competitive segregations onto the surface of Fe9Cr0.5Si alloy, open symbol: C_S_ = 0; solid symbol: C_S_ reaches the upper limit of sulphur solubility [93]. Reproduced with permission from Elsevier.

**Figure 16 materials-15-01331-f016:**
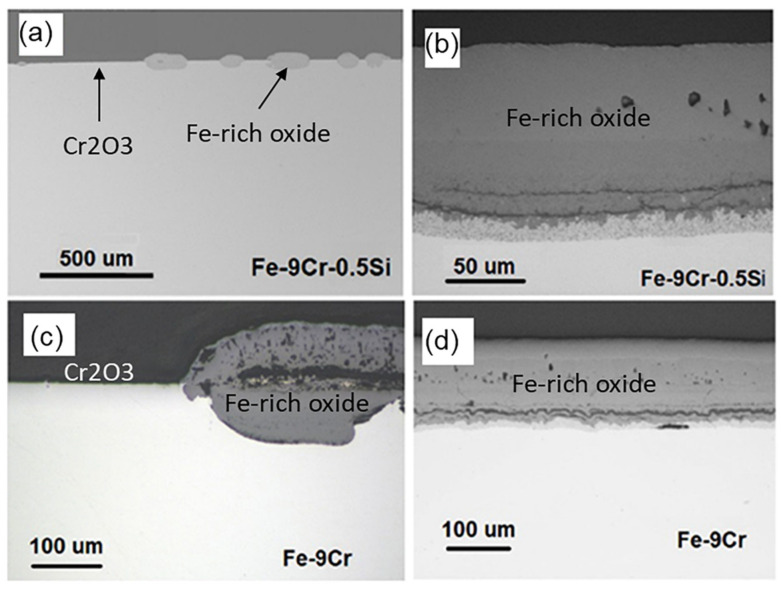
Cross-section overview of external scales for Fe9Cr and Fe9Cr0.5Si alloys exposed to Ar-20%CO_2_–20%H_2_O gas at 650 °C/1 bar with and without SO_2_: (**a**) Fe9Cr0.5Si, 1.0%SO_2_, 240 h; (**b**) Fe9Cr0.5Si, no SO_2_, 240 h; (**c**) Fe9Cr, 1.0%SO_2_, 500 h; (**d**) Fe9Cr, no SO_2_, 240 h [95]. Reproduced with permission from Elsevier.

**Figure 17 materials-15-01331-f017:**
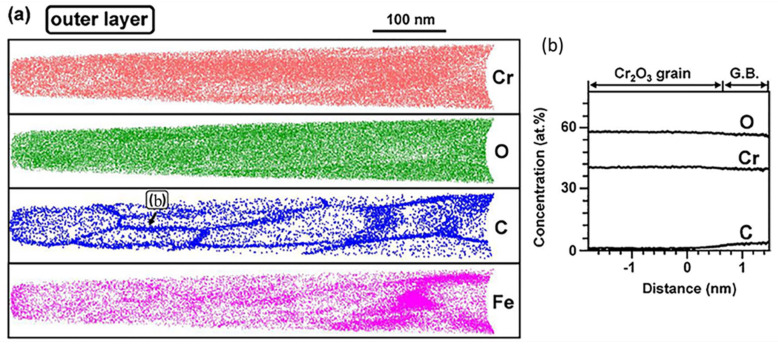
(**a**) Atom maps of slices through oxide tips of outer layer of the Cr_2_O_3_ scale on Fe20Cr reacted at 650 °C in Ar-20%O_2_ (24 h) and then in Ar-20%CO_2_ (70 h). (**b**) Proxigram of species in the outer layer with respect to the a C_2_O^2+^ 0.7 at. % isoconcentration surface from the area indicated by an arrow in (**a**) [101]. Reproduced with permission from Elsevier.

**Figure 18 materials-15-01331-f018:**
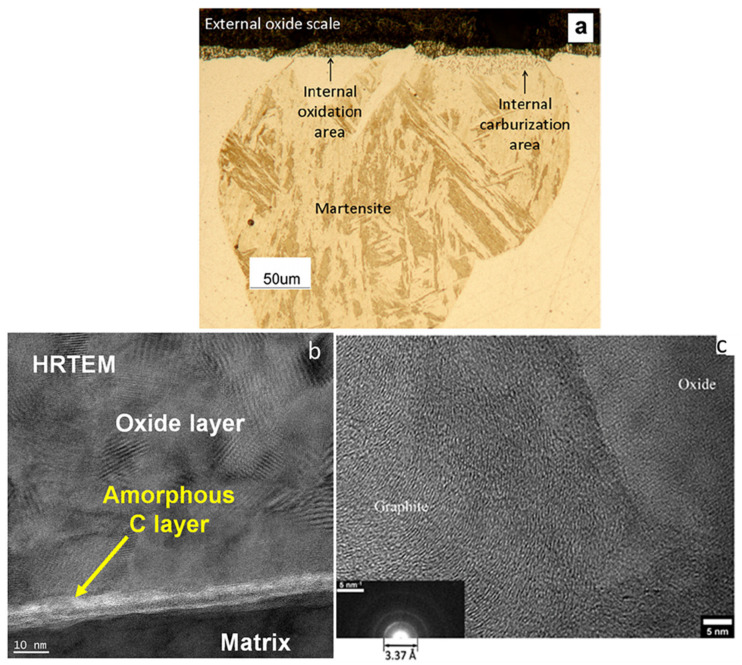
(**a**) Internal reaction products were observed in Fe9Cr alloy after exposure to 800 °C/1 bar Ar-20%CO_2_ for 20 h [33]. Reproduced with permission from Elsevier. HRTEM analyses results of oxide layer: (**b**) Ni14Cr binary alloy corroded in 600 °C/20 MPa S-CO_2_ for 200 h [97], reproduced with permission from Springer Nature. (**c**) Fe9Cr1Mo steel corroded in 600 °C/42 bar CO_2_ for 20,000 h [104]. Reproduced with permission from Elsevier.

**Figure 19 materials-15-01331-f019:**
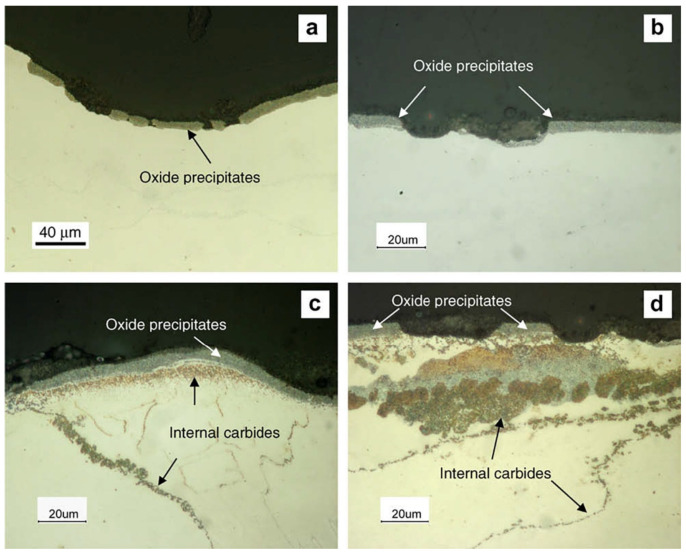
(**a**,**b**) Polished and (**c**,**d**) Murakami etched cross-sections of 304 austenitic steel after 141 cycles of reaction in CO/H_2_/H_2_O gas at 680 °C/1 bar [109]. Reproduced with permission from Elsevier.

**Figure 20 materials-15-01331-f020:**
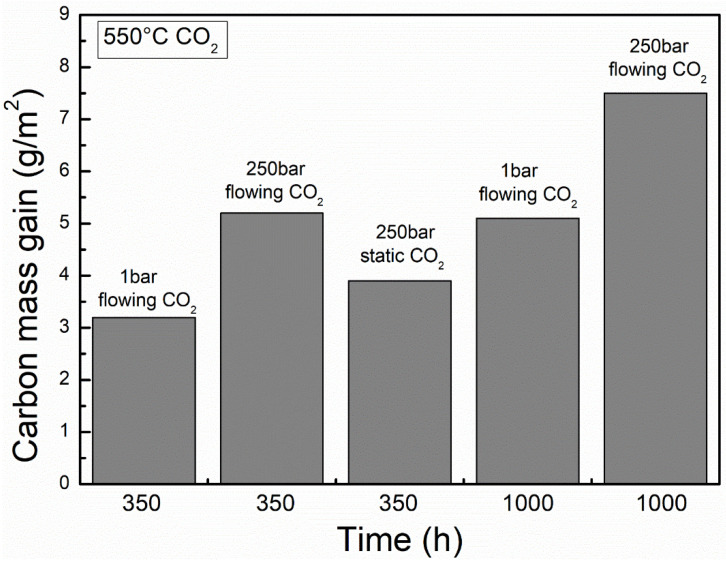
The carbon mass gain of 9Cr CEA (FMs T91) samples vary with exposure time and experimental conditions. Data were obtained from reference [17].

**Figure 21 materials-15-01331-f021:**
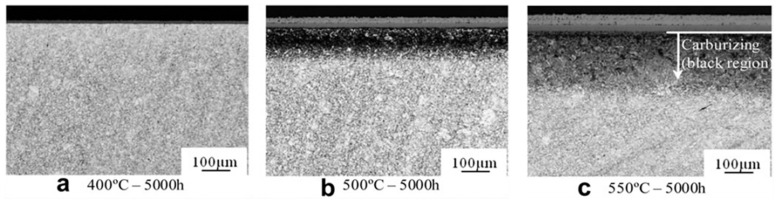
The microstructure of 12Cr FMs after 5000 h of exposure to (400–550 °C)/20 MPa CO_2_ gas [40]. Reproduced with permission from Elsevier.

**Figure 22 materials-15-01331-f022:**
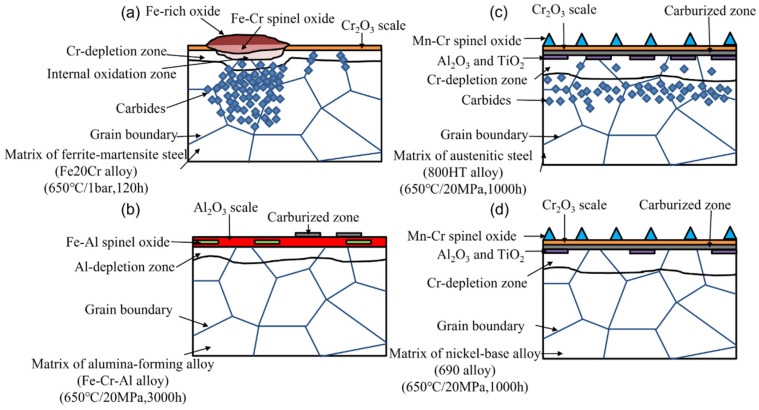
The schematic diagram of carburization of four typical alloys. (**a**) FMs Fe20Cr (reaction condition (RC): exposure to 650 °C/1 bar Ar-20%CO_2_ for 120 h) [33], (**b**) ODS Fe–Cr–Al (FMs) alloy (RC: exposure to 650 °C/20 MPa CO_2_ for 3000 h) [57], (**c**) austenitic steel 800HT (RC: exposure to 650 °C/20 MPa CO_2_ for 1000 h) [29] and (**d**) nickel-base alloy 690 (RC: exposure to 650 °C/20 MPa CO_2_ for 1000 h) [29]. The carbides above are caused by carburizing, and the precipitation of carbides after aging is not considered. The carburized zone represents amorphous carbon layer or graphite or free carbon.

**Figure 23 materials-15-01331-f023:**
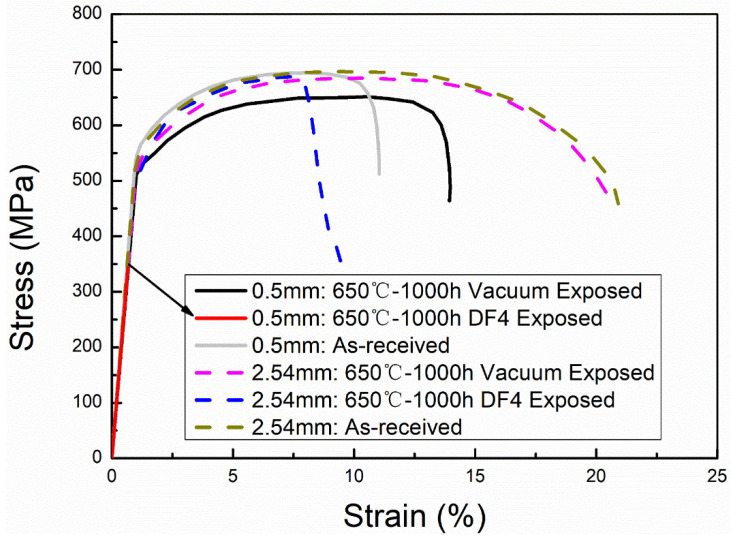
The engineering stress–strain curves obtained from the room temperature tensile testing on 2.54 mm/0.5 mm thick FMs P91 exposure to 650 °C vacuum or 650 °C/1 bar CO_2_ containing 4 vol % H_2_O and 1 vol % O_2_ (DF4) for 1000 h. The stress–strain curves of as-received samples are also shown. [34]. Reproduced with permission from Springer Nature.

**Figure 24 materials-15-01331-f024:**
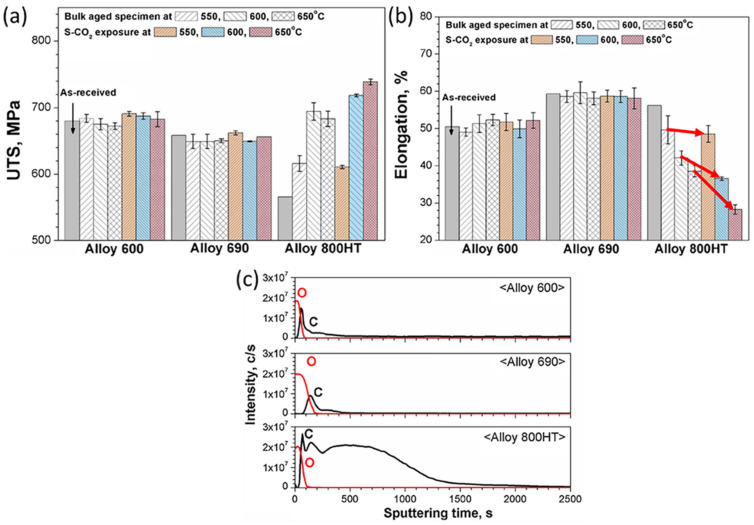
Results of room temperature mechanical properties for S-CO_2_ (20 MPa) exposed and bulk-aged samples at 550 °C, 600 °C and 650 °C for 1000 h compared to as-received conditions; (**a**) ultimate tensile strength and (**b**) elongation (arrows indicate the effect of exposure to S-CO_2_). (**c**) Results of SIMS depth profile of C and O for alloy 600, alloy 690 and alloy 800HT corroded at 600 °C for 1000 h [29]. Reproduced with permission from Elsevier.

**Figure 25 materials-15-01331-f025:**
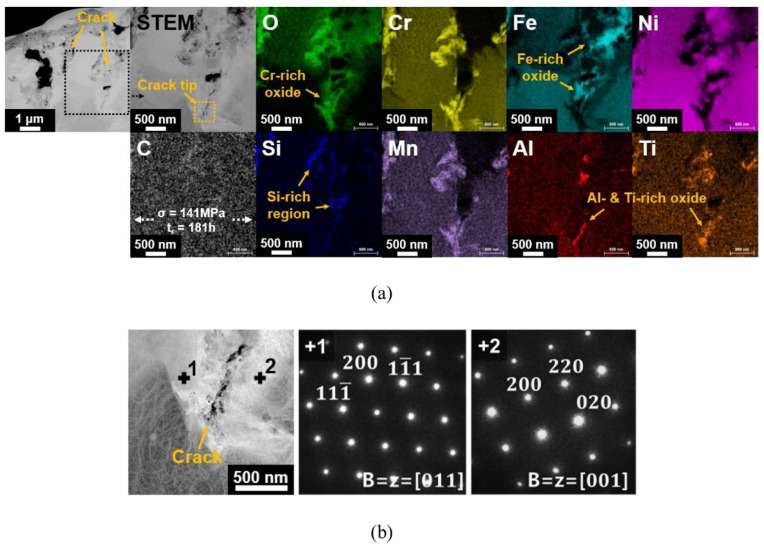
Results of TEM analyses of propagating crack region in S-CO_2_ crept specimen about 5 mm away from the fracture region; (**a**) STEM micrograph and EDS mapping image; (**b**) enlarged micrograph of area indicated in (**a**) and SAD patterns of grains adjacent to crack [129]. Reproduced with permission from Elsevier.

**Table 1 materials-15-01331-t001:** Results of tensile properties for FMs Grade 92 after the given exposure. Data were obtained from reference [20]. Reproduced with permission from Springer Nature.

	UTS (MPa)	0.2% YS (MPa)	Elongation (%)
Base: Unexposed	722 ± 4	562 ± 5	20.7 ± 0.5
Base: 450 °C CO_2_	785 ± 5	664 ± 4	18.1 ±4.3
Base: 550 °C CO_2_	766 ± 2	577 ± 3	13.4 ± 0.6
Base: 550 °C aging	735 ± 2	560 ± 1	19.9 ± 0.3
Welded: Unexposed	669 ± 1	509 ± 10	15.7 ± 0.2
Welded: 450 °C CO_2_	709 ± 3	593 ± 10	14.6 ± 0.2
Welded: 550 °C CO_2_	711 ± 3	518 ± 7	11.2 ± 0.6
Welded: 550 °C aging	674	496	15.1

**Table 2 materials-15-01331-t002:** Summary of creep rupture test results of alloy 600 in different environments. Data were obtained from reference [129]. Reproduced with permission from Elsevier.

Applied Stress (MPa)	Creep Rupture Life (h)		
	Air	CO_2_	S-CO_2_
135.73 (Air, CO_2_) 133.98 (S-CO_2_)	597.2	572.7	204.6
143.5 (Air, CO_2_) 141 (S-CO_2_)	464.2	334.5	181
151.33 (Air, CO_2_, S-CO_2_)	334.1	312.7	106.1
159.14 (Air, CO_2_) 158.26 (S-CO_2_)	201.6	189	104.3

## Data Availability

Not applicable.

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
