# Peer review of "Corrosion Behaviors of Heat-Resisting Alloys in High Temperature Carbon Dioxide"

_materials, 2022, doi:10.3390/ma15041331_

Round 1
Reviewer 1 Report
The paper provides an excellent overview of corrosion and oxidation at high temperatures and pressures of carbon dioxide. The article has good information and good writing. It is recommended that the following corrections be made:
- In Figure 1-3, descriptions of symbols such as blue and red signs and orange lines should be provided in the caption of the figure.
- Considering the importance of phase states and comparing temperatures and pressures in different cases, a phase diagram (temperature-pressure) should be presented in the introduction section.
Author Response
Dear reviewer,
Thanks very much for taking your time to review this manuscript. We really appreciate all your generous comments and suggestions! Please find my itemized responses in below and my revisions in the re-submitted files.

Reviewer 2 Report
After careful evaluation, I recommend for minor revision before publication in the Materials journal. The reason for my decession is as follows:
- The introduction section need to emphasize the importance of the work over the other.
- Comparative analysis with the standard alloys and materials need to be provided.
- Morphological changes and the physical composition need to be analysed using XRD, Raman and SEM analysis over the time of corrosion with the bare one.
Author Response

(The authors gave the same response as above.)

Reviewer 3 Report
The review work by Liujie Yang and coworkers describes the importance of materials selection for the powder generation by means of Supercritical carbon dioxide Brayton cycle. The authors covered the materials/alloys nature like the oxidation, carburization, and stress corrosion behavior under high temperature CO2. Finally, they provided some valuable suggestions for the selection of novel materials with structural properties applicable for the improvement of power generation in the supercritical carbon dioxide Brayton cycle.
Before accepting the work for publication, the following points needs to be covered.
- The manuscript needs English language to be corrected. Formatting the work with the help of professional English persons is strictly required.
- The authors covered only the three types of alloy (section 2) like Fe, steel, and Ni-based ones. It is good if the authors can improve this section by adding some more alloy types.
- Also, in section 2, the reason for selecting that particular type of alloy is also needed. The properties of alloys suitable for the power generation at high temperature conditions, like the morphology, energy states, thermal stability etc.
- The conclusions and scope of future research sections needs to be further improved by providing some valuable suggestions.

Author Response

(The authors gave the same response as above.)

Round 2
Reviewer 3 Report
I see significant improvement in the manuscript during the revision. Also, all my comments are being answered satisfactorily. Therefore the manuscript can be accepted for its publication.